# The vault associates with membranes in situ

Katharina Geißler [1,2,10], Jan Philipp Kreysing [1,2,10], Yuning Wang[1,2], Desislava Glushkova [1,2], Agnieszka Obarska-Kosinska [1], Patrick C. Hoffmann [1], Stefanie Böhm[1], Alexander Schmidt [3], Jakob Meier-Credo[4,5], Julian D. Langer [4,5,6], Gerhard Hummer [6,7,8] & Martin Beck [1,6,9] ✉

The eukaryotic vault particle is a giant ribonucleoprotein complex that assembles into an iconic barrel-like cage. Its cellular function has remained elusive despite extensive characterization. Using cryo-electron tomography of *Dictyostelium discoideum* cells, we define the distribution, structural states, and interaction landscape of vault particles in situ. Surprisingly, we detect a subpopulation of vault particles associated with the endoplasmic reticulum (ER) and nuclear envelope membranes. This association occurs at a defined barrel height of the vault particle. Membrane-associated particles appear to localize to patches of reduced membrane bilayer thickness and altered curvature. We further find that a fraction of vaults encloses 80S ribosomes in highly ordered orientations. These structural findings are further corroborated by proximity labeling experiments, which identify ER-resident proteins and numerous ribosomal components as vault particle interactors. The membrane-bound and ribosome-encapsulating vault populations that we uncover will direct future studies towards revealing vault function.

Cells are the basic building blocks of life. In highly organized structures, they harbor large molecular assemblies that carry out a multitude of critical functions. Many of these macromolecular machines, such as the ribosome or proteasome, are well understood with respect to composition, structure, and function. The vault ribonucleoprotein complex (vault), however, remains enigmatic despite its high abundance, massive size, and decades of research. The vault is a conserved eukaryotic 13 MDa assembly of proteins and small non-coding RNAs adopting a characteristic barrel shape. Since its serendipitous discovery in 1986 as a contaminant in rat liver vesicle preparations[1], it has been well characterized with respect to its molecular composition and structure. 78 copies of Major Vault Protein (MVP) are the main determinant of the particle shell[2]. Minor vault components are vault poly(-ADP-ribose) polymerase (vPARP/PARP4)[3], Telomerase-associated

protein 1 (TEP1)[4] and small, non-coding vault RNAs (vtRNAs)[1]. None of the minor vault components localizes exclusively to the vault, and to the best of our knowledge, a functional relationship in the context of the vault particle has not been described. The function of the vault, and in particular how it relates to the very unique cage-like structure, thus remains elusive.

The vault shell consists of 78 MVP monomers that assemble at polyribosomes by successive addition of dimers, formed by two nascent monomers interacting via their N-termini, to the growing structure[5]. MVP is a multidomain protein and contains Major Vault Protein-specific MVP domains, a Stomatin/Prohibitin/Flotillin/HflK/C family (SPFH) domain, an alpha-helical domain, and a short C-terminal disordered domain. While permanent membrane anchoring is a unifying feature of the SPFH protein family[6], the vault stands out for two

[1]Department of Molecular Sociology, Max Planck Institute of Biophysics, Frankfurt am Main, Germany. [2]IMPRS on Cellular Biophysics, Max Planck Institute of Biophysics, Frankfurt am Main, Germany. [3]Biozentrum, University of Basel, Basel, Switzerland. [4]Membrane Proteomics and Mass Spectrometry, Max Planck Institute of Biophysics, Frankfurt am Main, Germany. [5]Mass Spectrometry, Max Planck Institute for Brain Research, Frankfurt am Main, Germany. [6]Cluster of Excellence SubCellular Architecture of Life (SCALE), Goethe University, Frankfurt am Main, Germany. [7]Department of Theoretical Biophysics, Max Planck Institute of Biophysics, Frankfurt am Main, Germany. [8]Institute of Biophysics, Goethe University Frankfurt, Frankfurt am Main, Germany. [9]Institute of Biochemistry, Goethe University Frankfurt, Frankfurt am Main, Germany. [10]These authors contributed equally: Katharina Geißler, Jan Philipp Kreysing. ✉e-mail: martin.beck@biophys.mpg.de

reasons: First, membrane binding has not yet been reported. And second, due to the MVP domains, the vault forms a closed cage consisting of two halves, while SPFH complexes form 'half' cages that enclose membrane[7–11].

The presence and abundance of vaults in many eukaryotes (excluding only yeast, worms, insects, and plants[12]) together with the striking structural conservation[13] would imply a critical function. However, MVP deletion experiments in *Dictyostelium* amoebae that harbor two highly similar genes (MVPA and MVPB), so far revealed only a mild global phenotype of delayed growth under nutritional stress[14,15]. Tissue-specific effects in MVP knockdown mice[16–19] and differential MVP expression levels in human cells indicate that the vault may be involved in cell-type-specific functions. Descriptions of the vault's opening mechanisms[20–22] and the hollow, container-like structure furthermore suggest a potential 'shuttling' or 'carrier' function. To the best of our knowledge, such an endogenous function has, however, not yet been demonstrated.

In this study, we determine the vault's distribution in cryo-electron tomograms of *Dictyostelium discoideum* amoebae to delineate the subcellular context of these enigmatic particles. We analyze their in situ structures by subtomogram averaging. We find that a minor fraction associates with membranes of the nuclear envelope and the endoplasmic reticulum (ER). Moreover, by analyzing membrane thickness and curvature, we show that vaults enclose membrane patches of reduced thickness. We also identify a substantial fraction of vaults that encapsulate ribosomes with defined orientations, underscoring the idea that the vault may carry ribosomes as cargo[23]. Finally, we use proximity labeling to define the vault-associated proteome and thereby validate the findings of our structural analysis.

## Results

### Vault particles are highly abundant in *Dictyostelium discoideum* cells

Vault particles have previously been visualized in cells using cryo-electron tomography (cryo-ET) of human umbilical vein endothelial cells (HUVEC) and murine neurons[24,25], however, with limited throughput. Therefore, quantitative information on their spatial arrangement and distribution in living cells has remained scarce.

*Dictyostelium discoideum* is a eukaryotic model organism in which MVPs are highly expressed[26,27]. In a previous study, we acquired cryo-electron tomograms of *D. discoideum* perinuclear regions to assess nuclear adaptation to osmotic stress[28]. In this dataset, consisting of tomograms acquired in hyper-, hypo- or isotonic (control) conditions (EMPIAR-11845, EMPIAR-11943, EMPIAR-11944), we noted a high abundance of vault particles[23,24]. We therefore set out to determine their abundance and distribution across the three existing datasets by template matching (TM), where the tomograms are systematically examined for the unique structural signature of the vault in a cross-correlation-based approach.

We first generated a low-resolution template of the vault from the data itself by manually picking cytosolic vault-resembling particles with random starting orientations from ten *D. discoideum* tomograms. These 62 particles were then subjected to subtomogram averaging and alignment (STA) (see methods for details) to obtain an initial structure of the vault that was highly similar in size and shape to published maps[22,29–31]. Using this template, we performed TM using GAPSTOP[TM23,32,33] and extracted high-confidence matches based on cross-correlation score thresholding. The selected peaks were visually distinct and coincided with the typical structural features of the vault (Fig. 1a,b). After cleaning by manual inspection, we identified 999 particles in 318 tomograms.

We identified about three vaults per tomogram in control, hyper- or hypoosmotically shocked cells of the previously published perturbation experiment[28] (Fig. 1c). The majority of vaults localizes to the cytosol (Fig. 1d-f), which is consistent with previous reports[21,34]. No

discernible differences in vault morphology, spatial distribution, or particle number per tomogram were detected, and no irregularities were observed. Therefore, the datasets were merged to increase the overall particle count for subsequent analyses.

To further characterize structural features of the vault in situ, we performed STA of cytosolic vaults, resulting in a cryo-EM map at moderate resolution (29 Å, Fig. 1g, Supplementary Fig. 1). This structure reflected the characteristic barrel-like shape observed in the tomograms (Fig. 1d–f) and was in accordance with previously reported structures of human and rat vaults[22,29–31,35]. Similarly, the dimensions of *D. discoideum* vaults (approx. 75 nm × 40 nm, Fig. 1g) were comparable to those of rat (66-67 nm × 39–40 nm[29,30]) and human vaults (66-75 nm × 33-41 nm[22,24,31,35]). Thus, by exploiting an extensive state-of-the-art dataset, we were able to show that *Dictyostelium* vaults resemble those of metazoans in situ, both in terms of size and morphology.

### A minor fraction of vaults is associated with ER and nuclear envelope membranes

During manual inspection of TM-identified particles in the tomograms, we noted an unexpected subpopulation of vaults. Strikingly, a small fraction of particles was associated with membranes (14/999 particles, 1.4%. Figure 2a–c). Specifically, we observed binding to the endoplasmic reticulum (ER, Fig. 2d, f) and nuclear envelope (NE, Fig. 2e, g), with vaults being present at the respective cytosolic face of the organelle. We note that the original dataset targeted the nuclear periphery, such that this area is overrepresented.

Manual inspection of the membrane-bound particles revealed that they were always detectable in an upright position at the membrane. The part of the vault that is visible above the membrane accounted for approximately two-thirds of the particle, thus not representing a vault half-cage. In contrast, the part that is associated with the ER or NE (about one third) remained unresolved in the tomograms. Nonetheless, TM reliably identified such particles (Fig. 2a, b), indicating that structural characteristics of the cytosolic part, which protrudes from the membrane, are sufficiently preserved and not perturbed by membrane association.

### Vaults are associated with membranes at a specific barrel height

To compare the structure of membrane-associated vaults to their soluble counterparts, we performed STA of the membrane-bound subpopulation. The resulting subtomogram average (Fig. 2c) showed that vaults consistently associated with the membrane at their shoulder region at a defined barrel height. The structure of membrane-associated vault particles extended about 50 nm from the membrane plane and included the equatorial region in which two half vaults associate with each other.

While the part of the vault above the membrane remained intact, the cage-like density was not resolved underneath the bilayer density in the STA map, which is consistent with the primary data (compare Fig. 2c with Fig. 2d-g). Since both halves of the vault particle contain the alpha-helical domain, and truncation of the protein was not observed by Western blotting (Supplementary Fig. 2), this likely reflects incomplete visualization of the membrane-facing region of an otherwise intact vault cage and may imply that this part of the vault protein cage is structurally rearranged. The average also contained density that accounts for the membrane in a defined position, which was similarly observed in individual particles. These data suggested that the lower half of the vault specifically interacted with the membrane in its shoulder region, and that the membrane underneath was continuous (Fig. 2c–g).

### Membrane thickness and curvature are locally altered at vault's membrane association sites

Given this association of vault particles with membranes, we wondered if vaults coincide with patches of distinct membrane properties,

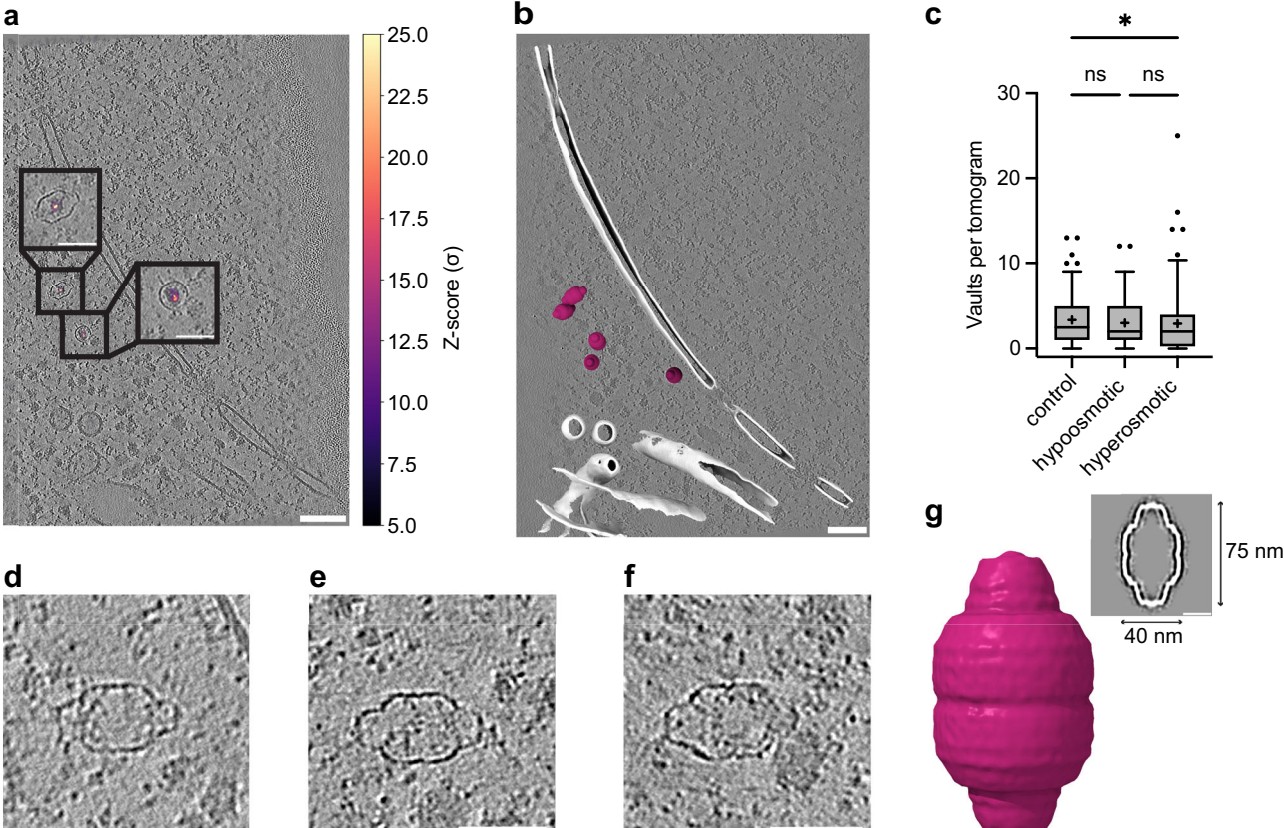

**Fig. 1 | Vaults are reliably detected by TM in *Dictyostelium discoideum* cryo electron tomograms. a** XY slice through an exemplifying tomogram of a *D. discoideum* cell. Vaults are highlighted in insets. TM peaks are shown superposed. **b** Visualization of cytosolic vaults in an exemplifying perinuclear *D. discoideum* cryo electron tomogram. White: segmented and isosurface-rendered membrane, berry: vaults. **c** Number of TM-identified vaults per tomogram (control $n$ = 446 in 132 tomograms, hyperosmotic $n$ = 330 in 112 tomograms, hypoosmotic $n$ = 223 in 74 tomograms. Mean (as +) and median (as lines) values are indicated, whiskers indicate 95-5 percentiles, and all points are displayed. Kruskal-Wallis test (alpha = 0.05) with Dunn's correction, $P$ = 0.0477 for control vs. hyperosmotic condition. **d**–**f**: Cytosolic vault particles. Image histograms and contrast are edited for improved visualization. **g** Subtomogram average of cytosolic vaults ($n$ = 985) shown isosurface-rendered and as a longitudinal slice through the STA map; dimensions are indicated. Scale bars: **a**, **b** 100 nm for overview and 50 nm for insets; **d**–**f** 50 nm; **g** 20 nm. Tomograms were denoised using cryoCARE[103] for visualization. Source data are provided as a Source Data file.

similarly to SPFH family proteins[6,36–40]. For this, we characterized membrane thickness and curvature as a proxy for local biophysical or functional membrane properties[41] at the site of the vault's association. We first manually extracted intensity profiles from longitudinal slices of our subtomogram average (Fig. 3a) and measured membrane thickness as either the distance between the inflection points on the profile or between the two minima (Fig. 3b), as previously described[41]. We observed a statistically significant reduction in membrane thickness in patches encaged by the vault (orange tones) compared to the surrounding membrane (blue tones, Fig. 3b, c).

To test whether this trend holds true at the individual subtomogram level, we again measured membrane thickness[41]. Consistent with our observations from the subtomogram average, we found that the membrane patch enclosed by the vault appears as a circular region of reduced thickness, while the site where the vault directly contacts the membrane shows increased thickness, visible as a 'ring' (Fig. 3d, bottom panel). The vault binding site also becomes apparent from the curvature analysis (Fig. 3d, mid panel). At the points where the vault shell associates with the membrane, the mean curvature is negative, i.e., the membrane is indented, indicating local perturbation of the bilayer.

### Vaults encapsulate ribosomes in highly specific orientations

We next analyzed the density inside the vault for both cytosolic and membrane-bound particles. Visually, they contained multiple smaller densities that appeared highly heterogeneous – consistent with previous reports[24]. We did not find any obvious repetitive or well-defined patterns across particles, making it difficult to classify and quantify reliably. However, further visual inspection of the tomograms suggested that a substantial fraction of all vaults contained ribosomes (exemplified in Fig. 4a-d), as had been proposed in a prior study based on a smaller data set[23]. To validate and quantify this observation, we utilized ribosome positions and orientations previously determined by STA[28,42] in this dataset and superpositioned them with the vault particle positions determined in this study. In total, 84 ribosomes were localized inside vaults. Of those, four ribosomes were encapsulated by membrane-bound vaults (4%, Fig. 4a, b), while 80 were enclosed by cytosolic vault particles (96%, Fig. 4c, d).

We next asked if 80S ribosomes or 60S ribosomal subunits were preferentially encapsulated. To investigate whether the ribosomes were intact, we performed TM of vault-encapsulated ribosomes with an 80S and a 60S ribosomal template. The vast majority of all targets (89%) (Fig. 4f) had higher cross-correlation scores for the 80S template compared to the 60S template. Even though this may be in part explained by the larger size of the 80S particle, our analysis does not support the hypothesis that 60S ribosomes are preferentially encapsulated.

We wondered if the encapsulated ribosomes exhibited a preferred orientation with respect to the vault and assessed this by comparing the previously assigned ribosome orientations[28,42] with the

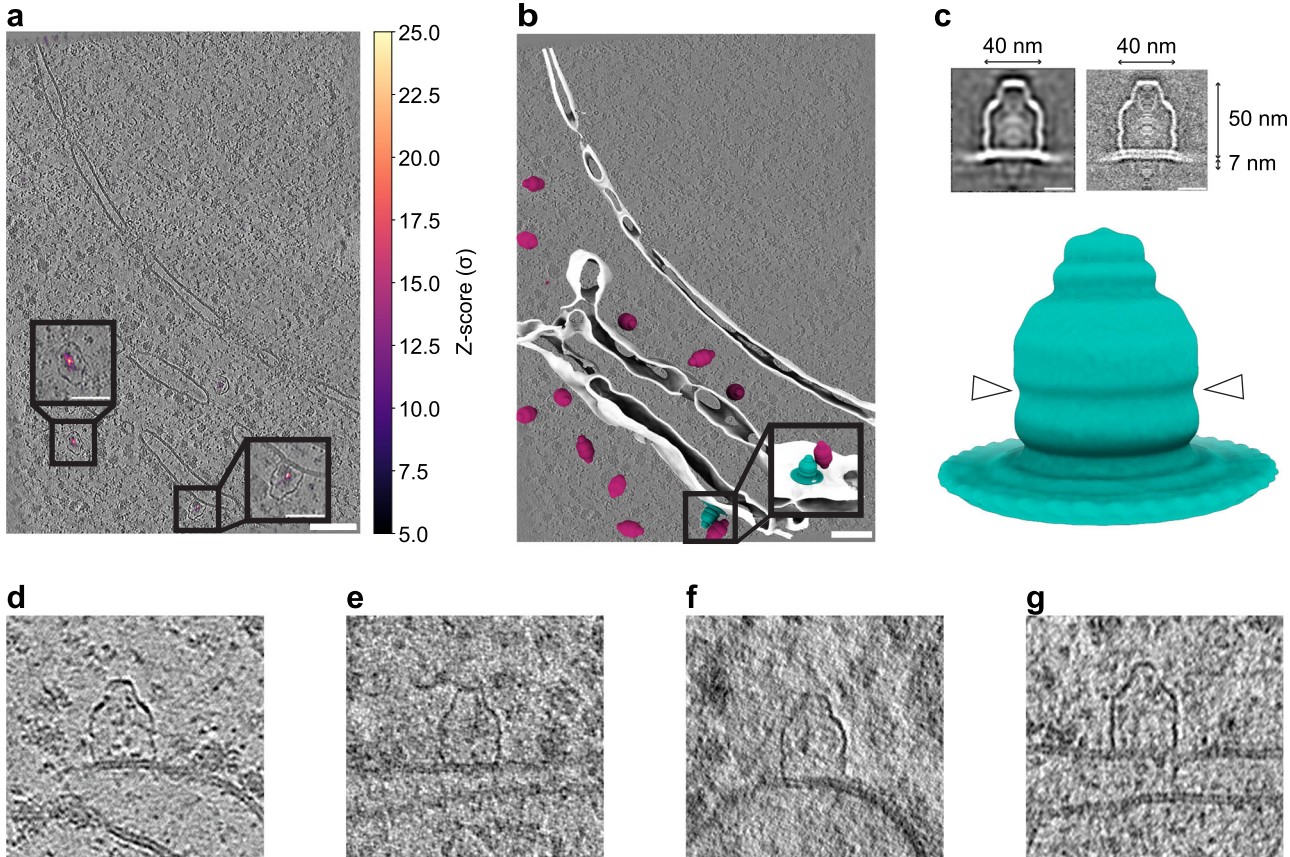

**Fig. 2 | A subfraction of vaults associates with membranes at a specific barrel height. a** XY slice through an exemplifying tomogram of a *D. discoideum* cell with TM peaks superimposed. **b** Visualization of cytosolic (berry) and membrane-associated (teal) vaults in the perinuclear region of an exemplifying *D. discoideum* cryo electron tomogram. White: isosurface-rendered ER and NE membranes. **c** Subtomogram average of membrane-associated vaults (*n* = 14) shown isosurface-rendered and as a longitudinal computational slice through the cryo-EM map; lowpass filtered to 6 nm in resolution (top left, bottom) und unfiltered (top right); dimensions are indicated. The particle number is too low for Fourier shell correlation analysis. Arrowheads indicate the equatorial mid-plane. **d–g** Examples of membrane-bound vault particles. Image histograms and contrast are edited for improved visualization. Scale bars: **a, b** 100 nm for overview and 50 nm for insets; **c** 20 nm; **d–g** 50 nm. Tomograms were denoised using cryoCARE[103] for visualization. A total of 14 membrane-associated vaults was observed. Source data are provided as a Source Data file.

vault orientations identified here. We indeed found that the orientation of the ribosome in vaults was non-random, but highly ordered (Fig. 4g, h). For cytosolic ribosome-vault pairs, we observed a narrow distribution of relative orientation angles between the ribosome and the vault, with their longitudinal axes approximately perpendicular (Fig. 4g). In this configuration, the ribosome exit tunnel faces the inner vault surface (Fig. 4j). This orientation is so invariant among the 80 observed cases that their average already yields a map in which the vault cage is partially resolved (Fig. 4i). Interestingly, the ribosome's long axis is oriented along the vault's short axis, an arrangement that seems spatially non-favorable.

The ribosome translocon complex (RTC) facilitates translocation of ER lumen-bound proteins co-translationally. As part of the RTC, the 80S ribosome is positioned with respect to the ER membrane in a specific orientation and distance to allow insertion of the nascent chain into the translocon channel[43]. In contrast to the vault-encapsulated ribosomes in the cytosol, three out of the four ribosomes observed inside membrane-bound vaults are positioned and oriented in a manner compatible with them being part of an RTC (hence referred to as RTC-like configuration). In this configuration, the longitudinal axes of ribosomes and vaults are roughly in register (Fig. 4h, k). We statistically compared the orientation of ribosomes contained within membrane-associated vault particles to those contained within cytosolic vault particles and found that it is significantly different (*p* < 0.0001 in a Fisher's exact test, see methods for detail). However,

the overall particle number is small and as such, this phenomenon should be further investigated in the future.

## ER- and ribosomal proteins are among the vault associated proteins

To independently test for interactions of vault particles with membranes and ribosomes in situ, we turned to TurboID-mediated proximity labeling. This method detects transient and low-abundant protein-protein interactions[44] by expressing a fusion of the protein of interest with a promiscuous biotin ligase, which biotinylates lysine residues of proteins within ~10 nm proximity[45]. Biotinylated interactors are isolated biochemically and subsequently identified and quantified by label-free mass spectrometry-based proteomics.

We generated *D. discoideum* strains that express the TurboID ligase fused to MVPA (*D. discoideum*-specific Major Vault Protein A) in addition to the endogenous MVPA, and a control line, where the ligase was expressed alone. To ensure that the TurboID labeling of MVPA does not interfere with the vault assembly, we assessed protein expression and complex assembly using subcellular fractionation in combination with size exclusion chromatography (Supplementary Fig. 2, Supplementary Methods). We found that the fusion protein, although low in abundance, was incorporated into the endogenous vault protein complex and sedimented with endogenous vault particles in subcellular fractionation experiments.

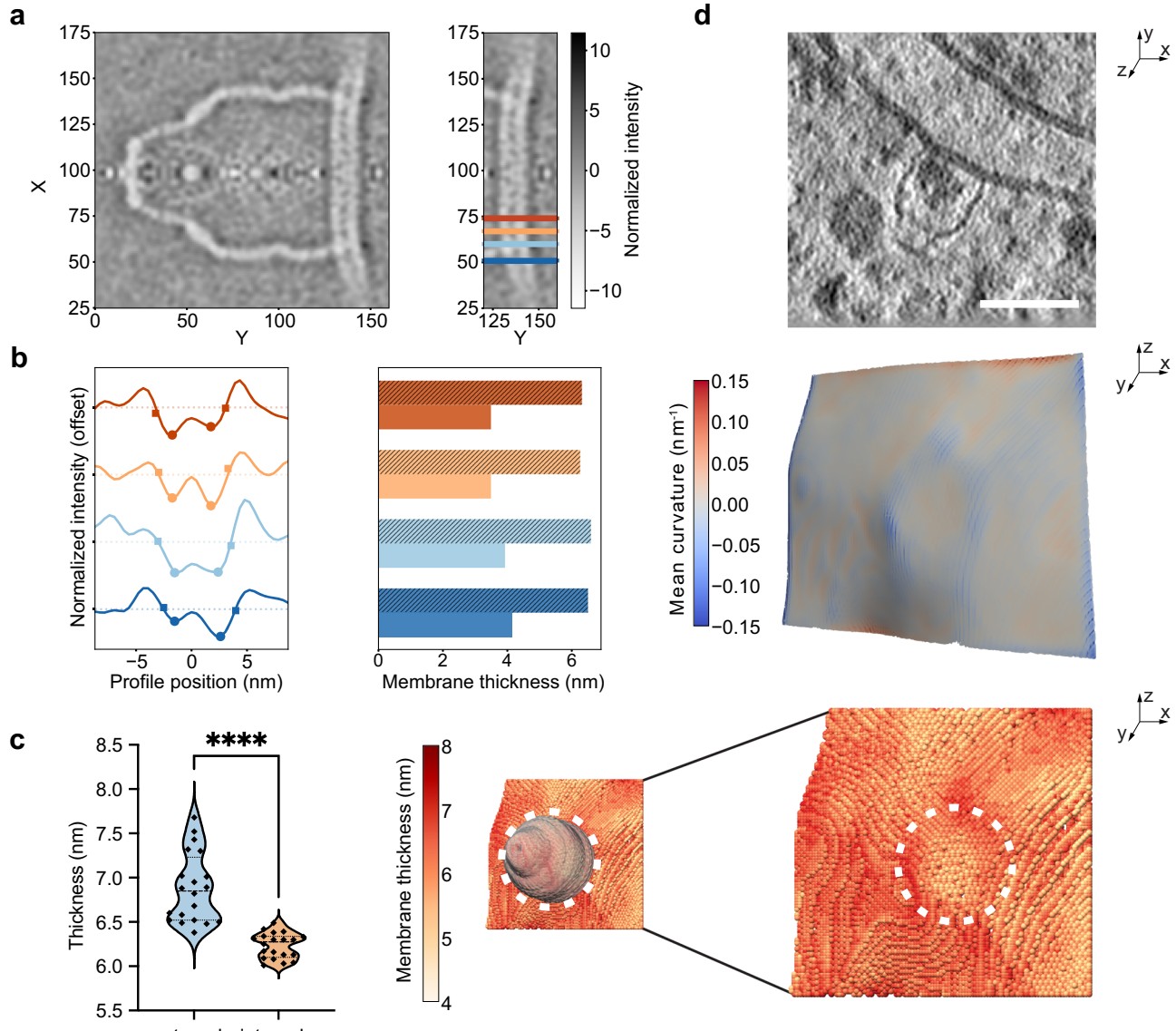

**Fig. 3 | Vaults localize to membrane patches of altered thickness and curvature.** **a** Central XY slice of the subtomogram average of membrane-associated vaults ($n = 14$) with z-score normalized color-coded intensity values (mean = 0, std = 1). Left: full slice; right: cropped slice with horizontal lines indicating sites of intensity profile extraction shown in (**b**). Blue tones: analysis points outside of the vault cage, orange tones: analysis points within the vault cage. **b** Left: exemplary intensity plots of the bilayer density with profiles centered on the maximum between the two leaflet minima (indicated as circles). The profile inflection points (squares) correspond to the maximum gradient positions between each minimum and its adjacent maximum. Right: corresponding membrane thickness measurements on the profiles, reported as distances between the inflection points (striped bars) or between the two minima (solid bars). Blue tones: analysis points outside of the vault cage, orange tones: analysis points within the vault cage. **c** Membrane thickness analysis of patches within the vault cage (orange) or its surrounding (blue) extracted from multiple slices of the subtomogram average map. Median thickness (inflection-point based): internal patch 6.28 nm, external patch: 6.85 nm, $N = 20$ each, U = 5, Two-tailed $p < 0.0001$ in a Mann-Whitney test. **d** Exemplifying visualization of the membrane thickness and curvature analysis workflow on individual sub-tomograms. Top: Example of a membrane-associated vault in a tomographic slice (XY view); mid: Membrane mesh (YZ view) color-coded by the mean curvature of each vertex. bottom: Membrane segmentation (YZ view), color-coded by local thickness with a model of the vault particle overlaid. Scale bar 50 nm. Source data are provided as a Source Data file.

Biotinylated proteins were affinity purified, digested into peptides and subjected to LC-MS/MS (Fig. 5a; see methods for details). To further validate the incorporation of the tagged MVPA into the vault complex, we first analyzed the detected peptides that were directly modified with biotin. For this, we generated homology models based on the human vault structure[31] for homooligomeric MVPA and MVPB and mapped the sites where TurboID auto-biotinylates the vault shell onto our homology models. As expected, biotinylated lysines on MVPA reside in the cap region (Fig. 5b), proximal to the C-terminus where the ligase was fused to the MVPA gene. Interestingly, MVPB, although not tagged, was also biotinylated (Fig. 5b). One may interpret these

findings such that the *D. discoideum* vaults, at least in part, form a heterooligomeric assembly in which MVPA and MVPB laterally interact with each other. This result further underscores that vaults incorporate the ligase into the native particle structure.

Quantitative analysis revealed 529 proteins that were significantly enriched in the MVPA-TurboID samples (q < 0.01), as compared to the control strain that samples the cytosol (Fig. 5a, d). Besides peptides derived from MVPA and MVPB, orthologues of known vault particle interactors were prominently identified, namely multiple poly(ADP-ribose)polymerases ("PARPs": Q54HY5, Q8I7C6, Q55GU8, Q54LJ4) and TEP1 (Q54CB5). Two further PARPs were identified (Q54E42 and

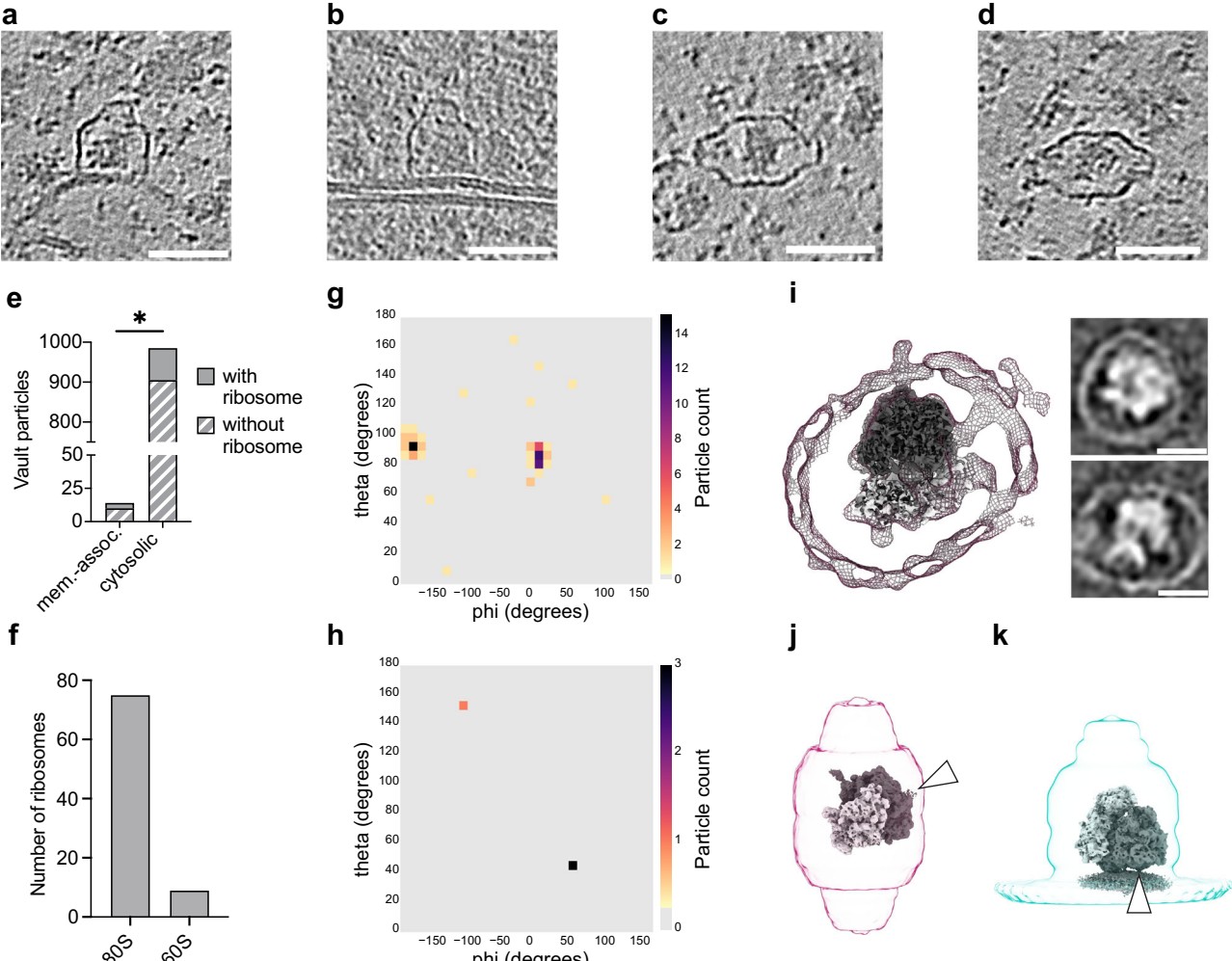

**Fig. 4 | Vaults encapsulate ribosomes in specific orientations. a–d** XY-slices through cryo electron tomograms of *D. discoideum* cells containing representative ribosome-containing vault particles, both cytosolic and membrane-associated. Scale bars 50 nm. Image histograms and contrast are edited for improved visualization. **e** Bar chart showing the number of cytosolic (80/985, 8%) and membrane-associated (4/14, -29%) vaults with (solid bar) or without encapsulated ribosomes (striped bar). *p* = 0.0242 in Fisher's exact test that compares encapsulation ratio in cytosolic vs. membrane-bound particle pairs, two-sided. **f** Bar graph depicting the result of TM detection of 80S ribosomes and 60S ribosomal subunits in vault-encapsulated particles. **g** Spherical histogram with distribution of relative orientations between cytosolic vault-encapsulated ribosomes and their enclosing vault (*n* = 80). **h** Same as **g** but between membrane-associated vaults and the respective

encapsulated ribosomes (*n* = 4). Ribosome orientations in g and h were sorted into angular bins representing RTC-like configuration and compared using a Fisher's exact test (two-sided), yielding statistical significance (*P* < 0.0001, see methods for details). **i** Central slice through cytosolic isosurface-rendered, vault-encapsulated ribosome STA map. Insets, right: Longitudinal computational slices through the STA map sliced through the equatorial vault axis (top) or through the long axis (bottom). Ribosomal subunits (EMD-15808) were fitted for visualization Scale bars: 20 nm. **j** Representative orientation of an exemplifying ribosome inside of a cytosolic vault. **k** Same as (**j**) but for membrane-associated vault. 60S in dark gray and 40S in light gray (EMD-15808, EMD-15809). Arrowheads indicate the ribosomal exit tunnel. Source data are provided as a Source Data file.

Q54XI2, peptide-counting-based z-score >3) but did not pass the filtering criteria for differential abundance analysis. These findings are consistent with previous studies of other organisms[3,4,35,46,47]. We further identified a large set of ribosomal proteins (69 out of 79 ribosomal proteins, 87%) of both the large and the small subunit as enriched in the set of vault-associated proteins as compared to plain cytosol (Fig. 5d, orange labels). We also identified several ER resident proteins as vault-enriched (Fig. 5d, teal labels). Those are either ER luminal or ER membrane proteins. These data support a model in which vaults bind to membranes and enclose ribosomes.

## Discussion
While the characteristic barrel-shape of the vault was reported decades ago[1], its cellular function remains enigmatic despite substantial research efforts. Knockout studies of vaults in mice and *Dictyostelium discoideum*[14,15,19] have shown no clear phenotypic effect in homeostatic

conditions. However, vault overexpression promotes survival of cancer cells and constitutes a common phenotype observed in multidrug-resistant cancer cell lines[48]. Therefore, the vault's function may only become crucial under specific cellular conditions and is thus challenging to address in experimental model systems. In line with this notion, both unanticipated states of vault particles that we report here, the membrane-bound and the ribosome-enclosing state, represent a small sub-population of all vault particles in steady-state unstressed wild-type cells.

The primary structure of monomeric MVP is organized into nine N-terminal Major Vault Protein (MVP) domains, followed by the <u>St</u>omatin/<u>P</u>rohibitin/<u>F</u>lotillin/<u>H</u>flK/C family (SPFH) domain, an alpha-helical domain, and finally, the C-terminus that is disordered (Fig. 6a, b). While the MVP domains are unique to the vault, all known SPFH domain-containing proteins share the following structural features: The SPFH domain is followed by the alpha-helical domain, and both

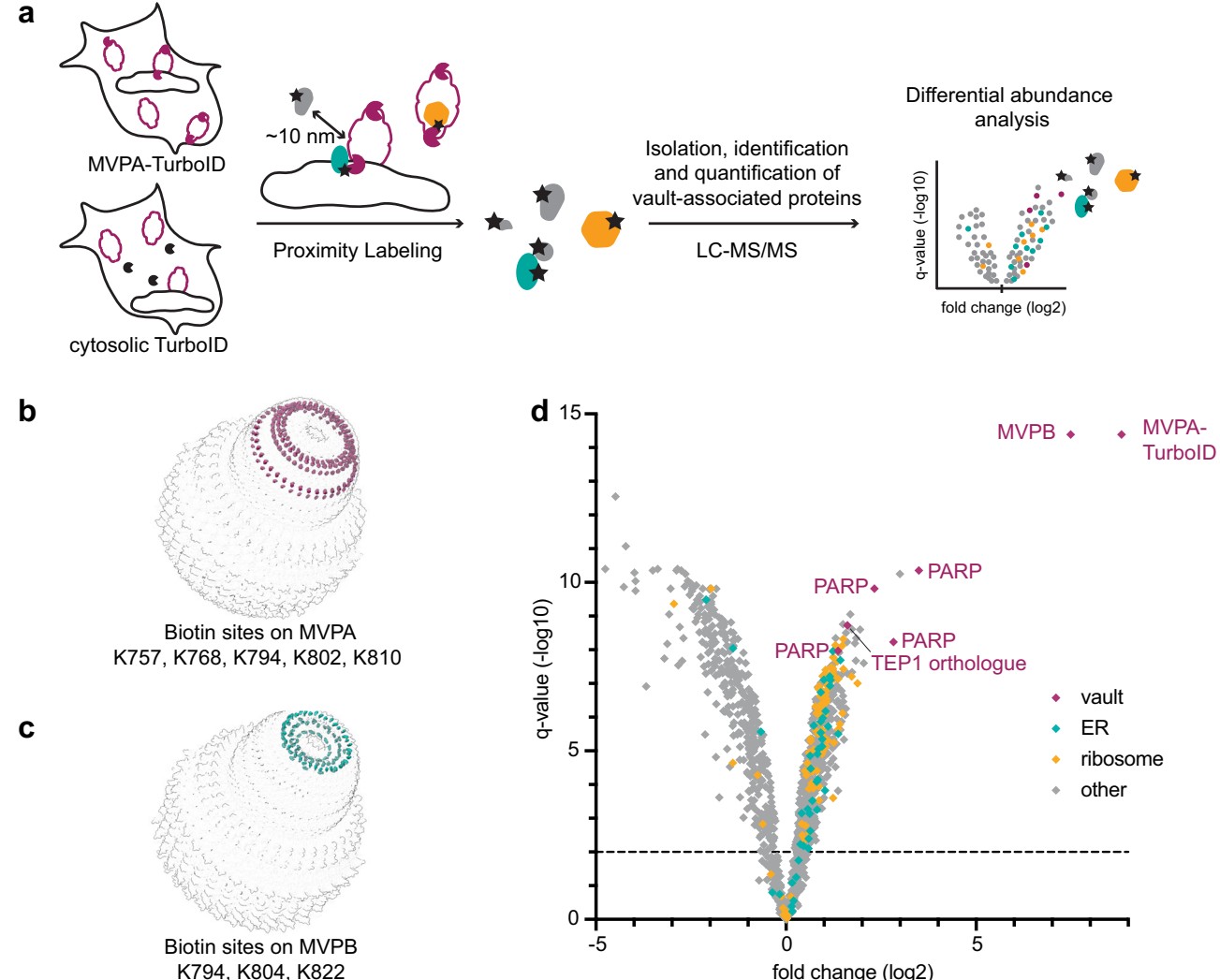

**Fig. 5 | Proximity labeling identifies ribosomal and ER proteins associated with vaults in cells. a** Schematic showing the proximity labeling approach to identify vault-associated proteins. b: Biotinylation sites mapped onto homology models of the *D. discoideum* vault confirm labeling radius and validate complex assembly. MVPA based vault homology models with biotin sites in berry. **c** Same as (**b**) but for MVPB based vault homology models with biotin sites in teal. **d** Volcano plot for differential abundance analysis of vault-associated proteins (right) vs. cytosolic control (left). Proteins colored according to association with GO terms: ribosome: orange, ER membrane and lumen: teal, vault components: berry. *N* = 4 or 5, FDR < 1%, DEA algorithm: limma (ebayes). Source data are provided as a Source Data file.

laterally assemble into giant higher homo- or heterooligomers reminiscent of 'half' cages[11,29,37,38,40,49] (Fig. 6c).

Canonical SPFH proteins bind membranes of various eukaryotic organelles or the *E. coli* inner membrane[7,37,38,40,50,51]. The respective membrane interactions are in one way or another mediated by motifs upstream of the SPFH domain. While Flotillin harbors hydrophobic residues and lipid-modified sidechains[11,40], other family members contain transmembrane or amphipathic helices that mediate membrane binding (Fig. 6b, c). Vaults contrast these canonical SPFH family members in several ways: (i) Membrane-associating motifs have not been described for the vault. (ii) Instead of membrane-associating motifs, the SPFH domain in the MVP monomer is preceded by multiple copies of the unique MVP domain. As a result, the vault cage is comparably large and has a characteristic shape (Fig. 6c). (iii) While other family members form 'half' cages, half vaults dimerize via the N-terminal MVP domain, thus forming a C2 symmetric 'full' cage. At last, and most strikingly, (iv) vaults bind to membranes in a fundamentally different way. In the vault, the N-terminal part of the SPFH domain is engaged with an MVP domain and is thus not free for membrane binding. Instead, the C-terminal part of the SPFH and the

alpha-helical domains point towards the membrane, thus effectively inverting the orientation of the cage with respect to the membrane compared to canonical SPFH family members.

We found that the association of vault particles with membranes occurred at a specific barrel height. Here, the alpha-helical domains forming one of the two vault caps remained unresolved in the EM maps, indicating that the otherwise intact cage may be rearranged from the SPFH domain onwards (Fig. 6c). In contrast, the lateral contacts formed by the MVP and SPFH domains between subunits were apparently largely preserved. The SPFH domain separates the alpha-helical domains from the MVP domains and provides the vault particle with a surface kink. One may speculate that this feature acts as a hinge between the MVP and the helix domains during membrane binding.

Intriguingly, vaults localize to thin membrane patches (Fig. 3). They may either bind to those specifically or, alternatively, segregate membrane patches of particular biophysical properties or composition. While direct membrane binding of vaults was not yet demonstrated, it has been reported that subpopulations localize to the cell surface of cancer cells[52] or the nuclear rim of rat fibroblasts[34]; that the vault may be recruited to plasma membrane patches in epithelial cells

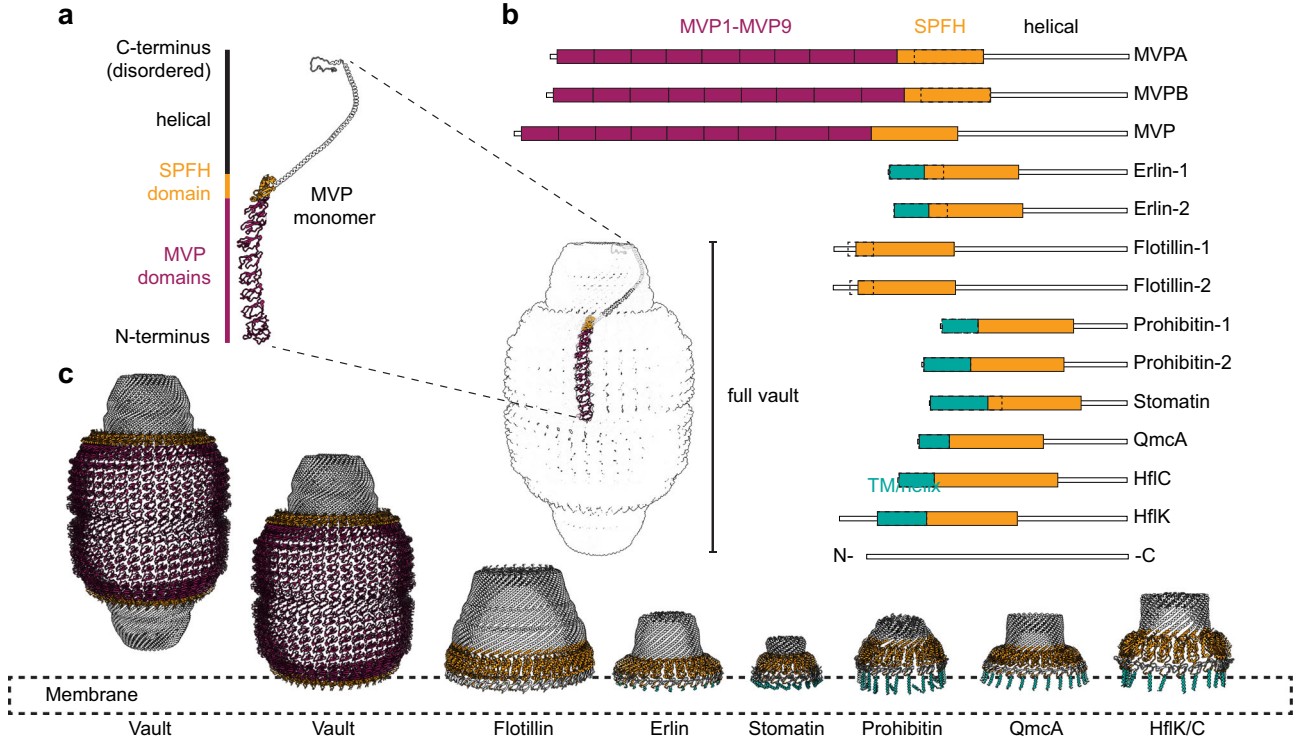

**Fig. 6 | Membrane binding mode of the vault in comparison with canonical SPFH family members. a** Representation of the MVP monomer domain architecture (top) and exemplary localization within the vault complex 78-mer (bottom). Model: PDB 4V60[30]. **b** Domain architecture of SPFH proteins. Berry: MVP domains, orange: SPFH domains, teal: transmembrane or membrane-associated helical domains, dashed line: approximate region of membrane association. **c** Side-by-side comparison of structural models and domain arrangements of protein cages (PDB 4V60[30], Flotillin 9BQ2[40], Erlin 9O9U[7], Stomatin 9OH9[50], Prohibitin 9O6S[51] QmcA 8Z5G[38], HflC/K 7WI3[37]). Orange: SPFH domains, teal: transmembrane or membrane associated helices, berry: MVP domains. Dashed line: Approximate membrane association region. The domains C-terminal of the SPFH domain in the vault PDB model were cropped to illustrate vault's membrane binding mode (second from left).

upon infection[53]; and that the vault comigrates with the membrane fraction of Xenopus egg extracts[54] and nuclear membrane receptors[55]. Our structural findings are consistent with these previous reports, and we propose a common model in which a minor fraction of vaults occurs in a membrane-bound form.

Our subtomogram averages of the vault, although moderate in resolution, overall agree well with previous in vitro structures and reflect the characteristic dome-like shape of the vault[29,30]. In contrast to previously reported in vitro structures, however, our in situ structure displays continuous density at its lids, consistent with the appearance of individual particles in tomograms (Figs. 1, 2). Furthermore, although half cages of the vault were previously observed under high salt conditions in vitro[13], we did not detect any half vault particles in our samples, neither by manual tomogram inspection nor by template matching. We, however, did identify membrane-associated vaults that exhibited an incomplete cage structure (Fig. 2), indicating that half or incomplete vaults likely did not escape our analysis per se, but rather that half vaults do not substantially contribute to the cellular population in steady-state *D. discoideum* cells. We also did not find particles supporting a role of the vault in cellular transport along microtubules[56], nor into the nucleus[34,57], as had previously been proposed. Although we cannot formally exclude such scenarios, they do not appear to constitute a frequent or abundant state in the samples we analyzed.

Our data do not allow us to ultimately complete the puzzle of the vault's function. A limitation of our study is the fact that the observed membrane-bound vault state constitutes a rare event. Nevertheless, the results of the proximity labeling experiments support the observations made by cryo-ET and speak for a membrane association. Given our observation that roughly one in ten vaults contains a ribosome

(Fig. 4), we furthermore speculate that vault particles may be connected to ER-associated ribosomal quality control (RQC) pathways. However, we mostly detect 80S ribosomes and not 60S ribosomal subunits inside of vaults, which would be inconsistent with the present model of ER-associated RQC that acts on 60S ribosomal subunits[58].

Alternatively, the vault could extract aggregated or dysfunctional proteins or lipids from the ER membrane. This would be in line with a role in membrane quality control similar to what has been proposed for Erlin and Prohibitin[59–61], two canonical SPFH family members. Similar to Erlin[7,62] and Prohibitin[63], vault particles localize to membrane patches of specific thickness and curvature. At this stage, it is unclear if membrane-bound vaults are in a dynamic equilibrium with the cytosolic form. If so, one could speculate that vault particles extract proteins from the ER and that the sequestered vault-cargo complex may then be subjected to autophagy. This would be in line with the finding that the vault directly interacts with the autophagy receptor NBR1 and is subsequently degraded by p62-mediated selective autophagy[46].

Future work is inarguably required to decipher the functional context of the vault's membrane association. Nonetheless, our study now provides distinct clues into which directions such experiments should be targeted to solve the vault function mystery. Due to the high structural similarity across organisms, we also expect that our findings will be relevant to other eukaryotes.

## Methods
### Template matching and initial template generation
First, a low-resolution vault template was generated from the control dataset (EMPIAR-11845)[28] itself by manually picking particles resembling vaults from ten tomograms with random starting orientations. After extraction of the 62 particles at 4x binning (8.704 Å/px) in 3D-

CTF corrected tomograms from novaCTF[64], subtomogram averaging and alignment was performed in novaSTA[65,66] to obtain an initial vault structure with imposed C39 symmetry. This low-resolution template was then used for template matching in GAPStop[TM23,32,33]. TM was performed on 4x binned, 3D-CTF corrected tomograms from the control, hyper- and hypoosmotic shock condition *D. discoideum* datasets (EMPIAR-11845, EMPIAR-11943, EMPIAR-11944[28]) with global angular sampling of 10 degrees while accounting for a C39 symmetry. The resulting constrained cross-correlation (CCC) peaks were then thresholded at a z-score of at least 18, and after removal of false positives by manual inspection, 999 vault particles were identified in 318 tomograms across the three datasets.

### Subtomogram averaging

After manual classification into cytosolic and membrane-associated vaults (985 and 14 particles, respectively), both particle sets were subjected to subtomogram averaging and alignment in novaSTA[65,66], with the cytosolic set being extracted at 4x binning and the membrane-associated at 2x binning from 3D-CTF corrected tomograms. Due to the low particle amount in the membrane-associated class, no FSC resolution by splitting the data into halfsets could be determined. Therefore, the resulting STA map is shown with a 60 Å lowpass filter. The 985 cytosolic vault particles could be split into halfsets before STA, resulting in a moderate $FSC_{0.143}$ resolution of 29 Å (see Supplementary Fig. 1 for FSC curve) as determined by Relion 3.1 postprocessing[67].

### Superposition analysis of ribosomes and vaults

To quantify how many ribosomes are encapsulated by vault particles, previously determined ribosome positions and orientations from all three *Dictyostelium discoideum* datasets[28,42] were correlated with the TM-detected vault particle set in a nearest neighbor analysis carried out with a function from the cryoCAT software package (*nnana.get_nn_stats*)[68]. A ribosome was only considered to be encapsulated if the center of the ribosome particle and the vault particle were no more than 10 nm apart. Visual inspection in the ChimeraX[69] plugin ArtiaX[70] confirmed that this selection of 84 ribosomes was indeed inside vault cages. Of those, four ribosomes are membrane-bound inside membrane-associated vaults, while 80 are cytosolic, enclosed by cytosolic vaults.

### Ribosome template matching

To assess whether the enclosed ribosomes are full 80S particles or rather 60S large subunits, template matching was employed again. Utilizing the already published *Dictyostelium discoideum* ribosome data[28,42] an 80S ribosome and 60S ribosomal subunit reference were separately aligned for one iteration in STOPGAP[33] against the 4x binned subtomograms of ribosome-containing vaults with 5 degree global angular sampling. This single round of alignment can be considered equivalent to TM on the same subtomograms. The CCC scores of each template against the same subtomogram were compared and an assignment as either 80S or 60S was made based on which CCC score was higher.

### Enclosed ribosome orientational analysis

To test whether the enclosed ribosomes have a preferred orientation inside vaults, we first simply averaged the extracted 4x binned subtomograms of the enclosed cytosolic ribosomes in novaSTA[65,66]. By using the existing orientations of the ribosomes from prior studies[28,42] a clear 80S ribosome encapsulated by a vault-like shell emerged from the averaging. The structure shown in Fig. 4i has been low-pass filtered to 50 Å as no resolution was determined by FSC.

To furthermore visualize how cytosolic ribosomes are oriented inside cytosolic vaults, a function in the cryoCAT package[68] (function: *visplot.plot_spherical_density*) was utilized to generate spherical histograms for both cytosolic ribosomes in cytosolic vaults and membrane-bound ribosomes in membrane-associated vaults.

For statistical analysis, we sorted the relative orientation of ribosomes inside of vaults into angular bins, which represent RTC-like or 'random' configuration, and asked if the RTC-like configuration could occur by chance. We performed a Fisher's exact test by comparing all ribosomes inside these angular bins for the membrane set (3 RTC-like, 1 not) and the cytosolic set (0 RTC-like, 80 not). We find a statistically significant ($p < 0.0001$, two-sided, Fisher's exact test) difference between the two different orientational distributions, as encoded in the respective $2 \times 2$ contingency table.

### Visualizations

To improve the visualization of tomographic slices in figures, denoising was applied to bin4 tomograms using cryoCARE[71]. Membranes were segmented for visualization (Figs. 1b, 2b) using MemBrain-seg[72]. Scatter, bar, box-whisker, or violin plots were generated using GraphPad Prism 10.4. Protein domains (Fig. 6) were visualized based on UniProt annotations and primary literature.

### Structural analysis

To visualize similarities of SPFH proteins, structural models were downloaded from the Protein Data Bank (PDB)[73] and visualized in ChimeraX[69]. SPFH proteins and Pfam[74] domain annotations were downloaded from the InterPro database[75] using identifier PF01145 (SPFH domain / Band 7 family).

### Homology models

Homology models of the *D. discoideum* vault were generated separately for the MVPA (P34118) and MVPB (P54659) proteins. For each protein, the corresponding AlphaFold prediction (AF-P34118-F1-v6 or AF-P54659-F1-v6)[76,77] was used to define the local domain structure, whereas the overall vault architecture was derived from human vault structures (PDB IDs 9R86 and 9R87)[31]. The human vault structures were first superposed in ChimeraX[78] to reconstruct complete chains. The *D. discoideum* AlphaFold models were then divided into fragments and aligned to the equivalent regions of the human template. Because 9R87 contains three slightly different chain conformations within the cap region, separate fragment-based templates were generated for each. These templates were used in SWISS-MODEL[79] to build the corresponding *D. discoideum* MVPA and MVPB homology models. Finally, the modeled monomers were assembled into complete *D. discoideum* vault particles by superposing them onto individual chains of the human vault structure using ChimeraX.

### Membrane thickness and curvature measurements

For membrane thickness analysis of the subtomogram average (STA), the STA volume was low-pass filtered to 20 Å resolution and z-score normalized with a mean = 0 and standard deviation = 1 (Fig. 3a). Intensity profiles were extracted along the membrane normal (y-axis) on multiple z-slices at coordinates corresponding to membrane regions outside the vault cage, at the intersection with the vault, and within the vault cage. Profile minima (the two membrane leaflets) were detected using the scipy.signal.find_peaks function on the inverted intensity signal and profiles were centered on the maximum between these two minima. Profile inflection points were determined as the positions where the intensity gradient reached a maximum between each of the two minima and its adjacent maximum. Two membrane thickness measurements were reported: the distance between the two minima and the distance between the two inflection points (Fig. 3b). Measurements were grouped by location, either as external or internal with respect to the vault cage. Statistical comparisons between the two groups were performed using the Mann-Whitney U-test in GraphPad Prism 10.4 (Fig. 3c).

For the per-subtomogram thickness and curvature analysis, 180 × 180 × 180 voxel volumes centered on membrane-bound vault particles were extracted from the original bin2 3D CTF-corrected tomograms and deconvolved using a Wiener-like filter implemented in the cryoCAT software package[68]. Membranes were segmented using MemBrain-seg with the pre-trained *MemBrain_seg_v10_alpha.ckpt* model and enabled connected component analysis[72]. Membrane thickness measurements followed a previously established workflow[41]. Briefly, the segmentations were used to generate a triangular mesh constrained to the segmentation boundaries with a 3D convolutional kernel. Each point on the mesh surface was assigned a coordinate and a normal vector. The two membrane leaflets were separated via principal component analysis (PCA), resulting in two distinct surfaces. Local membrane thickness was calculated as the Euclidean distance between paired nearest neighbor points on the opposing segmentation surfaces, reported in nm. One-to-one point matching was enforced to avoid duplicate measurements. The results were exported as membrane thickness motive lists compatible with ChimeraX[69] through the ArtiaX[70] plugin. Membrane thickness measurements were visualized in a color-coded representation, where color intensity reflects local changes in thickness (Fig. 3d).

The curvature of membranes was computed using a Python implementation of Rusinkiewicz's method for irregular triangular meshes[80]. Triangular meshes were generated from the MemBrain segmentations using marching cubes (scikit-image.measure.marching_cubes) with an iso-level of 0.5. Vertex normals were computed from face connectivity and refined using weighted neighbor averaging. Face curvature tensors were computed with least-squares fitting of normal differences across each triangle and were accumulated at vertices with Voronoi area weighting. Principal curvatures ($k_1$, $k_2$) and their directions were extracted with eigenvalue decomposition of the vertex curvature tensors. Mean curvature ($H = 0.5 \times (k_1 + k_2)$) and Gaussian curvature ($K = k_1 \times k_2$) were computed at each vertex. Mean curvature values on Fig. 3d are reported in units of $nm^{-1}$ and are visualized using ParaView[81].

## Plasmid generation

Plasmids for the generation of stable knock-ins of MVPA-TurboID fusion proteins at the *act5* locus were created using a previously reported *Dictyostelium* genetic engineering system[82]. Briefly, TurboID-(G4S)2-StrepII or MVPA-G2S-TurboID were cloned into pDM1513 or pDM1515 using NEBuilder HiFi DNA Assembly (New England Biolabs). ORFs of *D. discoideum* Major Vault Protein A (XP_646310.1) and TurboID were codon-optimized and obtained commercially. TurboID sequence was derived from the available *D. discoideum*-optimised BioID sequence[83] by introducing the described amino acid substitutions[84] using the most frequent *D. discoideum* codons. pDM1515 and pDM1513 were a gift from Rob Kay (Addgene plasmid # 109000 and #108998; RRID:Addgene_109000 and RRID:Addgene_108998).

## Generation of *Dictyostelium discoideum* cell lines

The axenic *D. discoideum* strains used in this study are derivatives of Ax2-214, obtained from the DictyBase stock center[85] (DBS0235534). Cells were electroporated with linearized plasmids using previously reported protocols[82] and clones were selected using 50 µg/mL Hygromycin B. Protein expression was confirmed using Western Blot with MVPA-specific antibody. Rabbit polyclonal antibodies raised against MVPA-specific peptides and affinity-purified using immobilized MVPA peptides were purchased from Davids Biotechnologie, Germany.

## Cell culture and proximity labeling

*D. discoideum* cells were grown axenically in HL5 medium without glucose (Formedium, supplied with 13.5 g/L glucose, 50 µg/mL ampicillin, and 50 µg/mL Hygromycin B for selection) at 21 °C/150 rpm as described previously[28]. 16 hours before the experiment, the medium was exchanged to chemically defined medium[86] (prepared in-house). After induction of biotinylation (15 minutes, 50 µM Biotin), cells were washed once with KK2 buffer (200 mM potassium phosphate, pH 6.6) and lysed chemically with protease inhibitor (Roche cOmplete, EDTA-free)-supplemented RIPA buffer (Thermo) for 10 minutes on ice. Lysates were cleared by centrifugation at 21000xg 4 °C for 5 minutes, flash frozen, and stored at −80 °C. The protein concentration was determined using the BCA assay (Pierce) according to the manufacturer's instructions, adjusted to 4 mg/mL with IP buffer (100 mM Tris-Cl, 150 mM NaCl, 1 mM EDTA (pH 8 at 25 °C), supplemented with PI) and 4 mg were used as input for pulldown of biotinylated proteins. Proximity labeling with subsequent sample preparation was done in 5 independent replicates.

## Affinity purification of biotinylated vault-associated proteins

Biotinylated proteins were isolated using 100 µL MagStrep StrepTactin Magnetic Beads slurry per sample (iba). Beads were equilibrated with IP buffer and incubated with 1 mL adjusted lysate for 2 hours with agitation and frequent resuspension in a deep well plate at 4 °C. Each sample was washed thrice with IP buffer with 0.1% SDS, and twice with IP buffer. After washing, beads were resuspended in 50 mM TEAB, pH 8.55, for MS sample preparation.

## Sample preparation for mass spectrometry

Cysteines were reduced with 20 mM DTT for 10 minutes at RT, followed by alkylation with 40 mM Chloroacetamide for 30 minutes in the dark at RT. Beads were washed three times with 50 mM TEAB pH 8.55, 1 µg rLysC (Promega, Mass Spec Grade) was added per well, and the digest was allowed to proceed for 16 hours at 37 °C with agitation. Subsequently, a peptide pre-fractionation step was performed similarly as described elsewhere[87]. Briefly, after on-bead digestion, non-biotinylated peptides were eluted from the beads first, followed by a second elution step that elutes biotinylated peptides (see below).

Nonbiotinylated ("bulk") peptides were eluted from the beads by vacuum-mediated filtration through a filter plate, followed by two further elutions with 50 mM TEAB. Eluted peptides were digested with 0.5 µg trypsin (Promega, Mass Spec Grade) for 3 hours at 37 °C and dried in vacuo.

Subsequently, biotinylated peptides were eluted from the beads twice by resuspension of the magnetic beads in 80% acetonitrile, 20% trifluoroacetic acid, followed by vacuum-mediated filtration through a filter plate. Eluted peptides were dried in vacuo, resuspended in 50 mM TEAB, and digested with 0.5 µg trypsin (Promega, Mass Spec Grade) for 3 hours at 37 °C and dried in vacuo.

All peptides were resuspended in 0.2% formic acid, desalted using C18-based StageTips as described elsewhere[88] and dried in vacuo.

## LC-MS/MS data acquisition

Peptides were solubilized by brief sonication in water/acetonitrile (ACN) (95/5, v/v), supplemented with 0.1% formic acid (FA) and subjected to LC-MS/MS analysis on a nanoElute 2 (Bruker) system, equipped with a C18 trap cartridge (5 mm × 0.3 mm ID, 5 µm particle size, ThermoFisher) and C18 analytical column (150 mm × 150 µm ID, 1.5 µm particle size, Bruker) coupled to a timsTOF HT mass spectrometer (Bruker) through a captive spray ion source with a 20 µm ZDV emitter.

**Bulk proteome.** Peptides were loaded onto the trap cartridge with 4x the inject volume of buffer A at a constant pressure of 500 bar. Separation was then carried out at 60 °C with a flow rate of 800 nL/min using a linear gradient from 2 to 38% B in 21 min, 38 to 95% B in 0.5 min, and constant 90 % B for 3.5 min with buffer A (0.1% FA in water) and buffer B (0.1% FA in acetonitrile).

Eluting peptides were analyzed in DIA-PASEF mode with a cycle time of 1.38 s and variable window sizes (12 ramps per cycle). DIA windows were optimized based on total *D. discoideum* lysate and pulldown DIA data using py_diAID[89,90]. Spectra were acquired over the mass range from 100-1400 m/z and a mobility window from 0.65-1.4 Vs/cm$^2$.

**Biotinylated proteome.** Peptides were loaded onto the trap cartridge with 4x the inject volume of buffer A at a constant pressure of 500 bar. Separation was then carried out at 60 °C with a flow rate of 800 nL/min using a linear gradient from 2 to 38% B in 21 min, followed by wash steps at 1 μl/min (38 to 65% B in 3.5 min and constant 90 % B for 5 min) with buffer A (0.1% FA in water) and buffer B (0.1% FA in acetonitrile).

Eluting peptides were analyzed in DDA-PASEF mode, with a cycle time of 1.17 s and 10 PASEF MS/MS scans. Spectra were acquired over the mass range from 100-1700 m/z and a mobility window from 0.6-1.6 Vs/cm2.

### Data analysis bulk proteome

Peptide identification and label-free quantification were performed in DIA-NN 2.2.0[91,92] against the *D. discoideum* Uniprot reference proteome (UP000002195, accessed 23.08.23, supplemented with sequences of the fusion proteins). First, a spectral library was predicted from the FASTA file. The database search space was restricted to tryptic peptides with a length of 7-35 amino acids, 2-4 charges, and a m/z range of 300-1800, allowing for up to two missed cleavages. Carbamidomethylation of Cysteine and N-term. Methionine excision was set as a fixed modification. Mass accuracy was set to 15 ppm, Match-Between-Runs, Unrelated Runs, and Protein Inference was enabled, and data normalization was disabled. The results were FDR-filtered (FDR < 0.01).

The resulting DIA-NN output results were filtered and subjected to differential abundance analysis using the MS-DAP pipeline (version 1.2.2)[93]. Peptides were filtered and normalized for each pairwise comparison ("contrast filtering") with the following settings: fraction_detect = 0.75; fraction_quant = 0.75; min_peptide_per_prot = 2; norm_algorithm = 'median'. The implemented eBayes (limma) algorithm was used for differential protein abundance analysis with an FDR threshold of $q < 0.01$. Thus, only proteins that were detected/quantified in 75% of samples (4/5 or 3/4 replicates) with at least 2 peptides per protein were used for differential abundance analysis. One replicate (MVPA-TurboID replicate #5) was excluded because it did not meet technical standards.

The MSDAP differential abundance analysis output table was annotated with Gene Ontology terms (downloaded from STRING V12[94] (organism ID:44689, protein.enrichment.terms) in R. Proteins were classified as follows: the vault: Manual assignment based on previous literature; ER GO:0005788 (ER lumen) AND GO:0005789 (ER membrane); ribosome GO:0005840. The manually annotated differential abundance data were subsequently exported and plotted using GraphPad Prism 10.4.

Proteins that likely represent orthologues of known vault-associated proteins were identified as follows: Poly(ADP-ribose)polymerases are already annotated as such in the reference proteomes. Among all identified PARPs, a variant annotated as PARP4/vaultPARP was identified in the data. For a putative TEP1 orthologue, the TEP1 sequence (human, Q99973) was subjected to BLAST[95] (via UniProt[96] release 2025_04) analysis against the *D. discoideum* genome. As the top hit, protein Q54CB5 was identified. Additionally, foldseek analysis[97] of human TEP1 revealed Q54CB5 as the top hit (E = 1, sequence identity 22.4%). Orthology information for TEP1 was retrieved from the OMA Browser (release July 2024[98]) and confirmed Q54CB5 as a likely *Dictyostelium* TEP1 orthologue in all 3 provided algorithms (pairwise orthologs, OMA Groups, and Hierarchical Orthologous Groups (HOGs)).

### Data analysis biotinylated proteome

Raw DDA-PASEF data were subjected to analysis in FragPipe (version 23), equipped with MSFragger (version 4.3[99–101]) for peptide matching based on the default LFQ workflow. Raw files were recalibrated, search parameters automatically optimised, and strict trypsin cleavage rules with a maximum of 2 missed cleavages were applied. Peptide length was restricted from 7 to 50, with a mass range of 500-5000 Da. Biotin (K, delta mass 226.07759) and N-terminal acetylation were added as variable modifications, with max. 2 variable modifications allowed on a peptide. Cysteine alkylation was added as a fixed modification. Percolator was used for PSM rescoring (min. prob. 0.5), and ProteinProphet was run for protein inference. Results were FDR filtered (protein and PSM level) with $q < 0.01$. Quantification was done in the IonQuant MS1 quantification module in LFQ mode. MBR was disabled, data were not normalized, and razor peptides were used for quantification. The *D. discoideum* Uniprot reference proteome (UP000002195, accessed 23.08.23, supplemented with sequences of the fusion proteins) was used with decoys and contaminants added.

For mapping of the biotinylation sites onto the vault homology models, the combined_site_report was used. In total, four biotinylated peptides were identified for MVPB and nine for MVPA, of which three mapped back to the TurboID part of the fusion protein and thus are not shown. Peptides with terminal modified lysines were excluded from the analysis. Thus, in total, 5 (MVPA) and 3 (MVPB) biotinylated sites are visualized. Visualizations were created in ChimeraX[69].

### Reporting summary

Further information on research design is available in the Nature Portfolio Reporting Summary linked to this article.

## Data availability

The STA map of the cytosolic *D. discoideum* vault is deposited in the EMDB with accession code EMD-56516. [https://www.ebi.ac.uk/emdb/EMD-56516]

Additional maps and homology models are available via Zenodo entry 18414818.

The cryo-ET datasets used in this study are available in the EMPIAR database with accession codes EMPIAR-11845, EMPIAR-11943 and EMPIAR-11944

Mass spectrometry proteomics data have been deposited to the ProteomeXchange Consortium in the PRIDE repository[102] with the dataset identifier PXD071329 Source data are provided with this paper.

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

## Acknowledgements

We thank Beata Turoňová for assistance with data processing and coding. We thank Michaela Müller-McNicoll for discussions. We thank Özkan Yildiz, Juan F. Castillo Hernandez, Thomas Hoffmann, and the Max Planck Computing and Data Facility for support with scientific computing. K.G., J.P.K., Y.W., and D.G. thank the International Max Planck Research School (IMPRS) on Cellular Biophysics. M.B. acknowledges funding from the Max Planck Society and the European Union (NPCvalve, project number 101054823 to M.B.).

## Author contributions

Study design: K.G, J.P.K., S.B., M.B. Data Acquisition: K.G., J.M.C. Data Analysis: K.G, J.P.K, Y.W., D.G., A.O.K., P.H. Formal Analysis: G.H. Visualization: K.G., J.P.K., D.G. Writing (original draft): K.G, M.B. Writing (Review & Editing): K.G, J.P.K, Y.W., D.G., A.O.K., S.B., P.H, J.M.C, J.D.L, A.S., G.H, M.B. Supervision: A.S., J.D.L., M.B.

## Funding

## Competing interests

The authors declare no competing interests.
