## [Transparent Peer Review file · Nature Communications]

The vault associates with membranes in situ

Corresponding Author: Professor Martin Beck

Version 0:

Reviewer comments:

Reviewer #1

(Remarks to the Author)

This manuscript, "Vault associates with membranes in situ", describes two sub-populations of vault particles in *Dictyostelium discoideum* amoeba. One associates to membranes of the nuclear envelope and the endoplasmic reticulum (ER), and the other encapsulates ribosomes.

Overall, the manuscript makes some interesting observations, however, there are several problems that need to be addressed before it can be recommended for publication. These problems are summarized below:

1. The authors (line 21 and the manuscript Title) redefine how the vault particle is referenced. They use "Vault" as a capitalized noun. This is confusing and unnecessary as vaults and the vault particle nomenclature has been used as such in the literature for the past 40 years. In addition, they blur the line between the singular and plural of vault. Even the title, "Vault associates with.." should be "The vault associates with ..." or "Vaults associate with ...".

2. An old hypothesis of vault assembly is presented (line 53-54, "each composed of 39 self-assembling MVP monomers"). This statement ignores a study published over ten years ago (their own reference 5, Mrazek, J. et al. ACS Nano 8, 11552–11559 (2014)) which demonstrated that vaults do not self-assemble, and they do not dimerize from half vaults. This polyribosome assembly model should be considered in light of the current data and the authors should either use their data to support or dispute the model.

3. The authors put considerable stock in the homology between the vault cap and the SPFH domain. This homology is distant and to set up the contrast between actual SPFH family members and vaults, is to set up a "straw dog" that is easy to refute.

4. Line 231, the inferred structural model of membrane-bound vaults, appears to be an issue and the validity of this structure may be in question.

The authors use the rat vault structure to interpret the membrane-binding region in *Dictyostelium*, which is a reasonable approach given the lack of a high-resolution Dicty vault model. However, Line 243-244, after identifying the membrane-binding region in rat (residues 504–507, rich in positive charges), the manuscript suggests that similar positively charged sequence segments were located in *Dictyostelium* and human vaults. If so, this comparison appears to be based solely on sequence similarity or residue numbering. Given that sequence position does not always correlate with spatial location across species, especially in large homologous assemblies, the authors should perform a structural alignment (they can use predicted models) to ensure that the corresponding residues in Dicty and human vaults are indeed located at the same region as the membrane-binding site observed in the rat structure. The authors need to clarify whether such an alignment was done? And if not, doing such an analysis would strengthen their argument.

5. The percentage of vaults containing ribosomes is less than 1 in 1,000. Could this be an artifact? Could vaults entrap ribosomes during vault biogenesis at a very low level? If the vault polyribosome assembly mechanism proposed by Mrazek, J. et al. is correct, could this extremely low occurrence of ribosomes in a vault be a "natural" artifact of the "accidental" trapping of a ribosome in a forming vault due to the crowded cytosolic milieu? If not a natural artifact, then the finding of a ribosome in a vault might be a significant clue to vault function. However, the 1 in 1000 occurrence, is thin evidence to support such a function.

Line 258, can the authors clarify how they calculated the ribosome occurrence in cytosol? This value is used to strengthen their argument that the occurrence of vaults with ribosomes is not a random accident.

6. Line 301 –The authors report observing four membrane-bound vaults that encapsulate ribosomes, and proceed to perform sub-tomogram averaging (STA) to propose that these ribosomes are in a "translation-competent" orientation. However, the small number of particles ($n = 4$) raises concerns about the statistical robustness of this conclusion. The authors should clarify how consistent the ribosome orientation was across these four particles? For instance, was any variability analysis performed to assess the degree of alignment or heterogeneity among them? Without evidence that these ribosomes are consistently oriented, and with such a limited dataset, the claim of "translation-competent" orientation may need to be framed more cautiously. To use the term "indicates" suggests strong data. At best "argues" is perhaps a better term and maybe still too strong.

7. Line 333 – this one report of vaults at lipid rafts published in 2007 has never been supported in later literature either by the publishing group or any other group.

8. Line 337 – This is likely not correct, as the reason vaults and CVs were co-purified was that they had an overlapping density and size. No "binding" association has ever been reported for vaults and CVs.

9. The authors reuse previously acquired cryo-electron tomograms from distinct experimental conditions (control, hyperosmotic, and hypoosmotic stress) for the current study. While reanalysis of existing datasets is a valid approach, it raises concerns when data from distinct conditions are combined for pooled analysis without sufficient justification. Could the authors clarify whether vault distribution patterns (e.g., membrane association or ribosome encapsulation) are significantly affected by the osmotic condition? Furthermore, although some statistical analysis appears in the figures, the manuscript lacks a clear description, interpretation, or discussion of these quantitative differences in the main text. A more detailed breakdown of the condition-specific results and how they support the conclusions would greatly improve clarity.

Reviewer #2

(Remarks to the Author)

Major vault is a giant oligomeric complex of unknown function in eukaryotic cells. Its main component, the major vault protein, oligomerize via 39-fold symmetry into a barrel-shaped closed structure, and two of these barrels further assemble to form a two-fold symmetric capsule in the cytosol. In their manuscript, the authors re-analyze existing cryo electron tomography datasets derived from *Dictyostelium discoideum* cells and apply template matching to identify a small (~1%) fraction of the major vault half capsule which assembles at the ER membrane and the nuclear envelope. Membrane-bound vault features minor structural rearrangements compared to the cytosolic form, while the membrane-binding region is not resolved suggesting structural changes upon membrane binding. The authors demonstrate that both cytosolic and membrane-bound vaults contain ribosomes. Based on their data, they suggest a function of membrane-bound vaults in membrane quality control or as ribosome carriers.

The manuscript starts with an interesting structural observation which could be potentially developed into a fascinating functional story. However, the authors do not follow up their structural observations so that the current manuscript remains purely descriptive. While the authors clearly demonstrate the value of in situ structural biology to observe interesting conformations/locations of cellular complexes, the paper also displays the limitations if such study is not paired with functional experiments. Thus, the authors propose a role of the major vault particles in quality control but from the current data, it could be similarly envisaged that the ER merely functions as an assembly platform for major vault protein and the half-capsules present assembly intermediates (which would be also an interesting story). Alternatively, one may propose that the ER-targeted half capsules are loading stations for cellular cargo, and many more scenarios could be imagined. *Dictyostelium* is a genetically amenable organism, so there are opportunities to carry out knock-out/knock-in experiments and express functional tagged versions of the major vault protein to study its cellular dynamics, obtain clues into its function and relate these experiments to the structural observations. Finally, even the described in situ structures could have likely been obtained at higher resolution resulting in more detailed structural insights into the conformational changes and the membrane interaction, if an optimized dataset with higher magnification had been recorded; there is no convincing argument to publish sub-optimal structural data. Much more work is required for this manuscript to become suitable for Nature Communication.

Reviewer #3

(Remarks to the Author)

This work encompasses a new evaluation of previously published electron cryotomography data after identification of rare membrane-associated and/or ribosome encapsulating Vaults within the dataset. The tomography and averaging approach appears to be technically sound, but the number of events identified is extremely small (14 for the membrane-associated Vaults, or ~1.5% of the total number of observed Vaults; 80 for the ribosome-encapsulated Vaults, or <10%) so the information gained from the structural analysis is very limited.

Some specific points:

1. Figure 1 panels E-I are all displayed at different apparent magnifications/levels of zoom, making comparison between the panels significantly more challenging.

2. The authors do not provide a figure of their proposed intact membrane-anchored Vault model (Figure 2D) docked into their subtomogram averaged map (Figure 2B), making it difficult for the reader to evaluate the fit of the model and interpret the apparent missing density away from the membrane (Figure 2). The authors also propose to deposit only the cytosolic Vault map from averaging in the EMDB, presumably due to issues around FSC/validation of the other maps due to the extremely limited particle numbers, which limits the ability for readers to perform this assessment on their own.

3. The authors make repeated reference to part of the Vault structure being 'dissolved' in the membrane-associated form. I find this wording troublesome, given that they state that they do not observe this region of density in their maps or tomograms (eg. line 218-219) and therefore cannot unequivocally state what has happened to it. It is possible that the helical region associates with the membrane, which could easily be obscured by the membrane itself at this resolution as well as CTF effects, is dynamic or is even cleaved. A better choice of wording here might be to simply use 'unresolved', especially as the symmetry-equivalent part of the Vault away from the membrane is also poorly resolved in the map due to the low particle numbers.

4. In the single slice shown in Figure 2B of the membrane-associated Vault subtomogram average, there appears to be two patches of density below the membrane, at spacings that by eye could be consistent with the 'dissolved' Vault density. These features are also apparent in the map the authors provided for review. Given the density is comparable to that observed at the opposite end of the Vault, it needs careful consideration and at first glance could appear to be somewhat in contradiction with the authors' conclusions.

5. I am hesitant about the interpretation of the connections between ribosomes and encapsulating vaults in Figure 3I, given the overall quality and SNR of the average which is apparent from the map provided to reviewers, with the vault itself being rather poorly resolved, most likely due to low particle numbers. As the authors propose to not deposit this map (I assume again due to FSC/validation issues) readers will be unable to make their own assessment of the interpretation. Likewise, interpreting a pattern in the orientation of the membrane-associated ribosomes from a sample of 4 individual events seems overly optimistic.

6. Given the extremely small number of membrane-associated/ribosome-encapsulating Vaults, a bigger cryoET dataset showing Vault and membrane/ribosome co-localisation/association would improve many of the issues described above and add significant weight to the robustness and importance of the authors' observations. An orthogonal experimental technique or model organisms could also help towards validating the observations.

Version 1:

Reviewer comments:

Reviewer #1

(Remarks to the Author)

I have re-reviewed the revised manuscript "The vault associates with membranes in situ" and I have read the detailed rebuttal letter describing how the authors have responded to all of the reviewer comments.

I think that the authors have done an excellent job responding and re-submitting a much more convincing manuscript that contains considerable new data and additional analysis of their previous data.

Although this manuscript does not solve the mystery of vault function, there are two significant findings in the manuscript that should contribute new insights to the field. I certainly agree with the authors statement in the rebuttal letter that their finding "that a minor fraction of vault particles associates with membranes at a defined barrel-height, and that it encapsulates ribosomes in preferred orientations in situ" is indeed novel and potentially important for defining additional functional experiments.

I have no additional criticisms.

Reviewer #2

(Remarks to the Author)

My assessment of this manuscript is not altered by the revision. The main finding of the manuscript, the identification and low-resolution structural characterization of membrane-bound vaults, is a noteworthy and surprising discovery, but without further functional or structural evidence, the relevance and the topology of the membrane-embedded vault remains enigmatic and the manuscript appears too preliminary for Nat Comm. I have also some concerns about the revision.

Figure 2: I indeed wrongly assumed in my first review that, similar to other SPFH members, a half capsule of the vault, not a full vault, binds to cellular membranes. This misunderstanding is partially caused by insufficient structural presentation and labelling, which has not been improved in the revised version: Based on the subtomogram average in Fig. 2C, it is not so obvious that a full vault particle is embedded in the membrane, and the reader should not wait until Fig. 6 to understand the main findings of the paper (see also point 2 of referee 3). To avoid confusion, Fig. 2 should already contain a structural model of membrane-bound vault in which the individual domains and the different regions of vault (e.g. cap, shoulder, etc) are clearly labelled and the molecules of the two hemi-capsules, as far as they are visible, are displayed in different colours.

Showing a half-capsule of the vault (previously in Fig. 2, now in Fig. 6B) next to the related half cages of other SPFH members is also not adding clarity in this regard.

Figure 2: With a full vault embedded in the membrane, its membrane-interaction mode is even more dazzling. Do the authors envision that the entire cap structure of the partially resolved hemi-capsule is post-translationally threaded through the membrane, with the MVP domains acting as transmembrane-spanning regions (as implied in the subtomogram averages of membrane-associated vaults in Fig. 2C, see point 4 of referee 3)? Such threading seems energetically costly, if not impossible. If the vault was such a membrane-traversing complex, how could then the membrane enter into the interior of the capsule? Alternatively, is the non-resolved half-capsule rather a peripheral membrane-binding disc, similar to caveolin, that can bind to membranes and be released without extraction of the transmembrane regions?

Fig. 3: Along the same line: The authors determine 'membrane thickness' within the vault-encapsulated membrane but a major portion of the vault protein present in this area is not resolved. How can the authors ascertain that the 'membrane' density in the interior of the vault does not represent the unresolved protein density (for example, in the form of a disc, see again for the caveolin structure) rather than the membrane; and that difference in 'membrane thickness' may be related to the presence of protein rather than differences in bilayer thickness? How should the vault select for short-chain lipids yielding a thinner membrane in its interior if not via protein-lipid interactions?

If the authors cannot provide further cell-based data to characterize the function of membrane-embedded vaults, they could attempt to characterize its membrane-binding determinants in terms of lipid composition and membrane curvature *in vitro*. Furthermore, they could reconstitute vault in suitable detergent or membranes and determine a higher resolution structure, in which its correct topology is determined and some additional insights into the formation of such complex are obtained. Such experiments would make this study suitable for *Nat. Comm.*

Further comments:

Line 280 et seqq.: Vault and ribosome translocon complex, orientation of the ribosomes within membrane-bound vault. I do not understand these arguments. Were the ribosome-encasing, membrane-bound vaults found on ER membranes where a translocon may be present, or rather at nuclear membranes? Is there any hint that the translocon is associated with vault complexes? To me, these ideas appear vastly speculative, and are also not supported by the low number of relevant particles; if at all, such suggestions should be included in the discussion.

Reviewer #3

(Remarks to the Author)

The changes the authors have made have significantly improved the manuscript. The addition of the mass spectrometry/proximity labeling in particular provides much needed orthogonal methods.

The authors have addressed the first set of points satisfactorily. A few small comments:

- 1) It would be useful to have some sort of indication of how many vaults showed non-ribosomal density inside the cavity, in order to put the ribosome-containing vaults in context
- 2) Some of the figures with tomogram slices suffer from uneven contrast/processing (in particular Fig 1 E-I, Fig 2 D-G) which detracts from the interpretability of the figures
- 3) In Figure 3A, on the right side panel, it is hard to see the horizontal coloured stripes on top of the orange to blue density colouring. Alternate colours or a greyscale density background would make this easier to see.
- 4) The particle count on Figure 4H should only show integer values - as its a count, non-integer values are confusing.

We want to thank all reviewers for their constructive feedback. In its content, the reviewers' critique is in large parts not unexpected to us. Before we go into a detailed point by point response, we would like to point out the following:

The vault has been discovered decades ago, but its function remains a mystery, despite extensive efforts. Many genetic and biochemical approaches, some of which were again explicitly suggested to us by the reviewers, e.g. to introduce knock-outs in *Dictyostelium*, have been already been tried, but yielded limited insights. The vault somehow eludes itself to genetics. It is abundant in certain cell types, highly inert, non-essential, but yet conserved in most eukaryotes. To understand its function has turned out very challenging. This is not our personal opinion, but the notion of an entire field¹.

In this study, we report our discovery that the vault binds to membranes. We were able to do so by analyzing an exceptionally large tomographic dataset with excessive computational power. Since then, we have been working very hard to functionally understand this finding and vault function in general. However, many of the experiments we conducted did not yield meaningful results, and are thus also not reported in our manuscript. For example, we have monitored vault localization using light microscopy under various stress conditions to explore if the membrane bound state can be populated, but it remains rare. We have done proteomics analysis of human MVP knockout cell lines, but did not detect any meaningful differences. We would be ready to include this data into the manuscript, but it would remain negative findings that will contribute very little to the conclusions.

We want to stress that, although we do deal with low abundant species, the frequency is actually not as low as it had been perceived by the reviewer(s) (e.g. reviewer #1 states 1 in 1000). The frequency of ribosome encapsulation is about 1 in 10; of membrane association 1 in 70; the union of both is only 1 in 250 – but we also do not draw any strong conclusions for the latter observation. The results we report are statistically significant, which is explained in further detail below.

We agree with the reviewers that our functional insights remain limited. We however have made substantial progress that we think deserves being reported, because it will help the field to consider these insights during future investigations. Specifically, the finding that a minor fraction of vault particles associates with membranes at a defined barrel-height, and that it encapsulates ribosomes in preferred orientations *in situ* is novel and striking.

We have extensively revised our manuscript and added additional data and analysis to address the reviewers' concerns. In brief, we have:

- Included mass spectrometry data that independently prove that vault interacts with both, ER-proteins and ribosomes (new Figure 5)
- Used an improved algorithm for membrane thickness analysis in cryo electron tomograms² and found that the membrane patch engaged by the vault has unique properties (new Figure 3)
- We included additional statistical analysis proving that our findings are significant, despite the low particle number
- We fundamentally restructured the results and discussion section for clarity, transparency and to remove too speculative arguments

We hope that the reviewers agree with us that our insights justify publication. Our detailed response follows below.

Reviewer #1 (Remarks to the Author):

This manuscript, “Vault associates with membranes *in situ*”, describes two sub-populations of vault particles in *Dictyostelium discoideum* amoeba. One associates to membranes of the nuclear envelope and the endoplasmic reticulum (ER), and the other encapsulates ribosomes.

Overall, the manuscript makes some interesting observations, however, there are several problems that need to be addressed before it can be recommended for publication. These problems are summarized below:

1. The authors (line 21 and the manuscript Title) redefine how the vault particle is referenced. They use “Vault” as a capitalized noun. This is confusing and unnecessary as vaults and the vault particle nomenclature has been used as such in the literature for the past 40 years. In addition, they blur the line between the singular and plural of vault. Even the title, “ Vault associates with.. “ should be “The vault associates with ...“ or “Vaults associate with ...”.

We thank the reviewer for pointing this out. We have carefully revised the text to make the naming consistent with the literature and changed the title to: “The vault associates with membranes *in situ*”.

2. An old hypothesis of vault assembly is presented (line 53-54, “each composed of 39 self-assembling MVP monomers”). This statement ignores a study published over ten years ago (their own reference 5, Mrazek, J. et al. ACS Nano 8, 11552–11559 (2014)) which demonstrated that vaults do not self-assemble, and they do not dimerize from half vaults. This polyribosome assembly model should be considered in light of the current data and the authors should either use their data to support or dispute the model.

We apologize if the wording of this sentence may have been misleading. In line with the study by Mrazek et al., we intended to refer to the fact that a newly translated MVP monomer at a polyribosome interacts with the neighboring MVP to form a dimer. We have revised the text as follows:

“The vault shell consists of 78 MVP monomers that assemble at polyribosomes by successive addition of dimers, formed by two nascent monomers interacting via their N-termini, to the growing structure⁵.”

We do not want to dispute the model mentioned, as we currently do not have data that would allow us to make conclusions on the mechanism of vault assembly. Initially, we had referred to the ribosome assembly model when discussing potential scenarios that could lead to the enclosure of ribosomes in vault (see point 5). However, we removed this section from our revised manuscript as it was too speculative and may have caused confusion. We note that the exit tunnel of the ribosome points to the equatorial plane of cytosolic vaults, which contains the N-

termini of MVP (new Fig. 4J). This is counterintuitive with a potential encapsulation during assembly in which the exit tunnel should rather point to the C-terminus (lid region).

3. The authors put considerable stock in the homology between the vault cap and the SPFH domain. This homology is distant and to set up the contrast between actual SPFH family members and vaults, is to set up a “straw dog” that is easy to refute.

We appreciate this comment and first want to clarify that we did not look for membrane binding vaults because of a possible relationship to SPFH proteins. It rather was a serendipitous discovery that sparked our curiosity about SPFH proteins.

We agree with the reviewer that the overall homology of SPFH family members and vaults is small in terms of sequence and therefore we would also not consider vault particles as ‘canonical’ SPFH family members. However, the fold of the SPFH domain itself is unique (as assessed by fold similarity searches using foldseek³) and conserved (see Response to reviewer figure 1). Likely due to this and additional similarities (characteristic C-terminal helices that interact laterally, oligomeric nature, large size of the cage-like assemblies in the MDa range), several recent studies⁴⁻⁷ have discussed the vault in the context of the SPFH family.

Response to reviewer figure 1. A: Overlay of SPFH domain alphafold predictions (Dictyostelium MVPA, MVPB, human MVP) or cryo EM structures (rat MVP, 7PKY¹⁹). Alignments were done in ChimeraX version 1.10 using the matchmaker function. B: Overlay of exemplary SPFH domain alphafold predictions (human Flotillin-1, Prohibitin-1, MVP SPFH). Alignments were done in ChimeraX version 1.10 using the matchmaker function.

Regardless of whether the vault should formally be considered an SPFH family member or not, we rather wanted to stress differences of the vault in structure and sequence as compared to other SPFH proteins. We have carefully revised the manuscript to make this clear to the reader. Specifically, we have moved the respective text to the discussion and modified it as follows:

“ ...Vaults contrast these canonical SPFH family members in several ways: (i) Membrane-associating motifs have not been described for the vault. (ii) Instead of membrane-associating motifs, the SPFH domain in the MVP monomer is preceded by

multiple copies of the unique MVP domain. As a result, the vault cage is comparably large and has a characteristic shape (Fig. 6C). (iii) While other family members form ‘half’ cages, half vaults dimerize via the N-terminal MVP domain, thus forming a C2 symmetric ‘full’ cage. At last, and most strikingly, (iv) vaults bind to membranes in a fundamentally different way. In the vault, the N-terminal part of SPFH is engaged with the MVP domain and is thus not free for membrane binding. Instead, the C-terminal part of the SPFH and the alpha-helical domains point towards the membrane, thus effectively inverting the orientation of the cage with respect to membrane compared to canonical SPFH family members.”

4. Line 231, the inferred structural model of membrane-bound vaults, appears to be an issue and the validity of this structure may be in question.

The authors use the rat vault structure to interpret the membrane-binding region in *Dictyostelium*, which is a reasonable approach given the lack of a high-resolution *Dicty* vault model. However, Line 243-244, after identifying the membrane-binding region in rat (residues 504–507, rich in positive charges), the manuscript suggests that similar positively charged sequence segments were located in *Dictyostelium* and human vaults. If so, this comparison appears to be based solely on sequence similarity or residue numbering. Given that sequence position does not always correlate with spatial location across species, especially in large homologous assemblies, the authors should perform a structural alignment (they can use predicted models) to ensure that the corresponding residues in *Dicty* and human vaults are indeed located at the same region as the membrane-binding site observed in the rat structure. The authors need to clarify whether such an alignment was done? And if not, doing such an analysis would strengthen their argument.

We agree with the reviewer that sequence similarity or residue numbers are not sufficient. We have done the respective structural alignment and found that the charged surface is conserved in *D. discoideum* and other MVPs (*DdMVPA* K489, K491, R492; *DdMVPB* K504, K507, R508; *HsMVP* R504, R506 and R507).

However, we also realized that the charges that are very prominent on the inside surface, are not apparent on the outside surface in our homology models of membrane-associated vault. Now one could argue that this may be explained with alpha helical and cap domains that are somehow rearranged and may participate in membrane binding. This is however speculative. We therefore decided to remove this aspect from the manuscript.

5. The percentage of vaults containing ribosomes is less than 1 in 1,000. Could this be an artifact?

We realized that the panel in the (former) figure 3F, which had shown the fraction of vaults encapsulating a ribosome, may have been misleading. We want to clarify that 84 out of 999 vaults encapsulate ribosomes and that this is a statistically significant observation. To avoid any further misunderstanding, we have revised the entire figure and removed the respective panel. The actual numbers of ribosomes in vaults are stated in the text and the overall numbers are represented in the new Fig. 4E.

Furthermore, we now include additional mass spectrometry data in the revised manuscript that underscore the association of MVP with ribosomal proteins in situ (new Figure 5).

Could vaults entrap ribosomes during vault biogenesis at a very low level? If the vault polyribosome assembly mechanism proposed by Mrazek, J. et al. is correct, could this extremely low occurrence of ribosomes in a vault be a “natural” artifact of the “accidental” trapping of a ribosome in a forming vault due to the crowded cytosolic milieu? If not a natural artifact, then the finding of a ribosome in a vault might be a significant clue to vault function. However, the 1 in 1000 occurrence, is thin evidence to support such a function. Line 258, can the authors clarify how they calculated the ribosome occurrence in cytosol? This value is used to strengthen their argument that the occurrence of vaults with ribosomes is not a random accident.

Please see response to point 2 above. We apologize for this misunderstanding and want to clarify that the occurrence of a ribosome inside of vault is actually 1:10. In the revised version of the manuscript, we additionally provide independent support for the ribosome association of vault, as we identified a large set of ribosomal proteins in our TurboID proximity labelling experiment (new Fig. 5D).

We have previously determined the position of all ribosomes in our *D. discoideum* tomograms^{8,9}. The calculations were done by superposing the coordinates and orientations of ribosomes from the previous study with the coordinates and orientations of vault particles from this study.

It also occurred to us that this could occur during vault biogenesis. We however did not manage to generate any additional evidence for this, e.g. there are no assembly intermediates apparent in our tomograms. As pointed out above, the exit tunnel of the ribosome is in a specific position pointing towards the N-terminus of MVP. We therefore do not speculate about this in our manuscript any longer.

6. Line 301 –The authors report observing four membrane-bound vaults that encapsulate ribosomes, and proceed to perform sub-tomogram averaging (STA) to propose that these ribosomes are in a “translation-competent” orientation. However, the small number of particles (n = 4) raises concerns about the statistical robustness of this conclusion.

As pointed out above, the orientation of ribosomes was determined previously^{8,9} and performed for a large number of ribosomes jointly by subtomogram averaging. The overall confidence of the angular assignment in this data set is very high.

We realize that our use of the term translation-competent may have been somewhat ambiguous. What we wanted to express was the fact the orientation of the ribosomes relative to the ER membrane, and their distance from it, matches what would be expected if these ribosomes were part of ribosome-translocon complexes (RTCs) inserting nascent proteins into the ER-lumen. In the revised manuscript, we now refer to this configuration as RTC-like.

We agree with the reviewer that n is small, but this finding is statistically significant – meaning that the likelihood of finding three out of four ribosomes in the same

orientation and position by chance is very small. In the revised manuscript, we have tested if the orientation of ribosomes within cytosolic vault particles is significantly different as compared to those within membrane associated vault particles. To this end, we sorted the orientation of ribosomes inside of vaults into angular bins and asked if the membrane facing ribosome orientation (“RTC-like”) could occur by chance. We performed a Fisher’s exact test by comparing all ribosomes inside these angular bins for the membrane set (3 ribosomes in RTC-like configuration, 1 not, refer to the response below how we define the translation competent state.) and the cytosolic set (0 RTC-like configuration, 80 not). We find a statistically significant ($p < 0.0001$) difference between the two different orientational distributions.

Alternatively, one can ask if the orientation of ribosomes within membrane-associated vaults can occur by chance. To statistically test for this, we generated a set of random ribosome orientations based on non-encapsulated ribosome-vault pairs from our data. More specifically, we generated set of orientations based on a random subset of 500 cytosolic, non-encapsulated ribosomes with respect to arbitrary cytosolic vault particles *in silico*. We sorted this set into angular bins as described above. We statistically compared the relative orientation of ribosomes to vault this set (2 compatible with RTC-like configuration, 498 not) to the relative orientation of the 4 ribosomes inside of membrane-associated vaults with respect to their enclosing vault particle (3 ribosomes in RTC-like configuration, 1 not). The Fisher’s exact test revealed a statistically significant difference ($p < 0.0001$). This test is not shown in the manuscript and only reported here for comparison.

We revised the manuscript to make this more transparent. The respective analysis is shown in new Fig. 4 G,H. We think it does make sense to report this observation. In the revised discussion we are very cautious to not draw any conclusions from it.

The authors should clarify how consistent the ribosome orientation was across these four particles? For instance, was any variability analysis performed to assess the degree of alignment or heterogeneity among them? Without evidence that these ribosomes are consistently oriented, and with such a limited dataset, the claim of "translation-competent" orientation may need to be framed more cautiously. To use the term “indicates” suggests strong data. At best “argues” is perhaps a better term and maybe still too strong.

Regarding consistency, three ribosomes are virtually identical, while one of the four is rotated (see response to reviewer figure 2).

As noted above, our phrasing was not ideal. We intended to refer to the similarity of this orientation to the RTC in both the distance from the membrane, and the orientation of the ribosome. Only the latter was statistically tested, but as evident from response to reviewer figure 2 above, ribosome position and orientation perfectly coincide in those 3 ribosomes.

Response to reviewer figure 2: Visualization of membrane associated vault particles containing ribosomes. Subtomogram averages are shown isosurface-rendered to visualize the superposition of the ribosomes within membrane-associated vault particles (n=4). While the three arrangements on the left are reminiscent of ribosomes in RTC-like configuration, the one to the very right is rotated.

However, since it is unlikely that tRNAs can access vault-encapsulated ribosomes, the choice of the term ‘translation-competent’ is somewhat misleading. We rephrased the text as follows:

“In contrast to the vault-encapsulated ribosomes in the cytosol, three out of the four ribosomes observed inside membrane-bound vaults are positioned and oriented in a manner compatible with them being part of an RTC (hence referred to as RTC-like configuration). In this configuration, the longitudinal axes of ribosomes and vaults are roughly in register (Fig. 4H, K).”

7. Line 333 – this one report of vaults at lipid rafts published in 2007 has never been supported in later literature either by the publishing group or any other group.

We appreciate this comment. However, the respective manuscript is published and part of the existing literature. We therefore think it does make sense to at least mention it in the part of the discussion that is meant to reconcile our findings in the context of previous studies. The respective passage is rather cautiously phrased: *“While direct membrane binding of vaults was not yet demonstrated, it has been reported ... that the vault may be recruited to plasma membrane patches in epithelial cells upon infection⁵³ ... ”*

This sentence does not even reiterate the major conclusion of this manuscript (*“Host Resistance to Lung Infection Mediated by Major Vault Protein in Epithelial Cells”*) that the reviewer may be concerned about. Neither does it mention lipid rafts in the revised version, a concept that meanwhile also has been challenged.

8. Line 337 – This is likely not correct, as the reason vaults and CVs were co-purified was that they had an overlapping density and size. No “binding” association has ever been reported for vaults and CVs.

We have removed this statement, as it was speculative and alternative explanations for this observation are conceivable, as the reviewer notes.

9. The authors reuse previously acquired cryo-electron tomograms from distinct experimental conditions (control, hyperosmotic, and hypoosmotic stress) for the current study. While reanalysis of existing datasets is a valid approach, it raises concerns when data from distinct conditions are combined for pooled analysis

without sufficient justification. Could the authors clarify whether vault distribution patterns (e.g., membrane association or ribosome encapsulation) are significantly affected by the osmotic condition? Furthermore, although some statistical analysis appears in the figures, the manuscript lacks a clear description, interpretation, or discussion of these quantitative differences in the main text. A more detailed breakdown of the condition-specific results and how they support the conclusions would greatly improve clarity.

We agree that the use of these three datasets should be transparent and therefore had included an analysis in Figure 1C, establishing that vault abundance and structural states are only marginally affected by those treatments. We therefore pooled the data obtained of the three different conditions to increase the size of the dataset and added this information to the manuscript:

“We identified about three vaults per tomogram in untreated, hyper- or hypoosmotically shocked cells of the previously published perturbation experiment²⁷ (Fig. 1C). The majority of vaults localizes to the cytosol (Fig. 1 D-F), which is consistent with previous reports^{20,33}. No discernible differences in vault morphology, spatial distribution, or particle number per tomogram were detected, and no irregularities were observed. Therefore, the datasets were merged to increase the overall particle count for subsequent analyses”

Reviewer #2 (Remarks to the Author):

Major vault is a giant oligomeric complex of unknown function in eukaryotic cells. Its main component, the major vault protein, oligomerize via 39-fold symmetry into a barrel-shaped closed structure, and two of these barrels further assemble to form a two-fold symmetric capsule in the cytosol. In their manuscript, the authors re-analyze existing cryo electron tomography datasets derived from *Dictyostelium discoideum* cells and apply template matching to identify a small (~1%) fraction of the major vault half capsule which assembles at the ER membrane and the nuclear envelope. Membrane-bound vault features minor structural rearrangements compared to the cytosolic form, while the membrane-binding region is not resolved suggesting structural changes upon membrane binding. The authors demonstrate that both cytosolic and membrane-bound vaults contain ribosomes. Based on their data, they suggest a function of membrane-bound vaults in membrane quality control or as ribosome carriers.

The manuscript starts with an interesting structural observation which could be potentially developed into a fascinating functional story. However, the authors do not follow up their structural observations so that the current manuscript remains purely descriptive. While the authors clearly demonstrate the value of in situ structural biology to observe interesting conformations/locations of cellular complexes, the paper also displays the limitations if such study is not paired with functional experiments. Thus, the authors propose a role of the major vault particles in quality control but from the current data, it could be similarly envisaged that the ER merely functions as an assembly platform for major vault protein and the half-capsules present assembly intermediates (which would be also an interesting story). Alternatively, one may propose that the ER-targeted half capsules are loading stations for cellular cargo, and many more scenarios could be imagined. *Dictyostelium* is a genetically amenable organism, so there are opportunities to carry out knock-out/knock-in experiments and express functional tagged versions of the major vault protein to study its cellular dynamics, obtain clues into its function and relate these experiments to the structural observations. Finally, even the described in situ structures could have likely been obtained at higher resolution resulting in more detailed structural insights into the conformational changes and the membrane interaction, if an optimized dataset with higher magnification had been recorded; there is no convincing argument to publish sub-optimal structural data. Much more work is required for this manuscript to become suitable for Nature Communication.

We thank the reviewer for appreciating our in situ structural findings. Indeed, the observation that vaults bind membranes is exciting given the fact that the function of the vault remains a conundrum despite decades of research. We agree that our work at this stage does not ultimately answer that question. We however respectfully disagree with the overall assessment of our study. Likely similar to many other groups, we have performed numerous experiments trying to determine the cellular function of vaults, however, none of these have provided conclusive results (yet). As pointed out in the common response to all reviewers above, we are convinced that the additional insights that our study provides are a major step towards answering these questions. Importantly, in the revised manuscript we now include additional mass spectrometry data (new Figure 5), which provides orthogonal experimental evidence for our conclusions, thus going beyond a structural observation.

We therefore hope that the reviewer agrees that our findings deserve publication and that the scientific field can then build on these findings to eventually solve the vault mystery.

Regarding the specific points of the reviewer:

With respect to the proposed alternative functional models the reviewer suggests, we would first like to clarify that we do not observe “half-capsules” or half vaults like those that have previously been proposed¹³, in our tomograms. Furthermore, the membrane bound form of vault does not represent ‘half vaults’. In our scenario, both halves are present, but one is incompletely represented in the averages because the helical and disordered parts, which would extend beyond the membrane, are not resolved. Since vault particles are known to assemble co-translationally in solution¹⁴ and no cofactors are required¹⁵ and particles are even formed recombinantly in *E. coli*¹⁶, we find it extremely unlikely that membrane-bound particles represent assembly intermediates – and this interpretation would contradict present models of vault biogenesis¹⁴.

Knock-out experiments in *Dictyostelium*, which the reviewer suggests, have been done^{17,18} and revealed little but a delayed growth upon nutrient depletion. We therefore instead performed a screen for vault localization using light microscopy under various conditions of cellular stress, which technically worked, but did not yield any results towards elucidating the function of vaults. We also analyzed the proteome of human MVP knock-out cell lines, again without any meaningful results. Given this very limited insight, we did not include those data into the present study. At this stage we are uncertain if we can resolve this puzzle in a timely manner and thus want to make the community aware of our findings up to this point.

At last, we want to clarify that the very same tomographic data set has revealed high resolution structures of ribosomes⁸ and is state of the art in terms of size and sampling. We agree that the resolution of the vault is moderate, primarily because the individual subunits do not rotationally register. It is however not clear if that were to happen at higher magnification, and even if so, what more in terms of function could we expect to observe at higher resolution? We expect that the fact that a fraction of vaults binds to membrane by exposing its SPFH domain would remain. The exact lipids it binds would however remain unresolved. At the same time, a better pixel size would reduce both field of view and throughput.

Reviewer #3 (Remarks to the Author):

This work encompasses a new evaluation of previously published electron cryotomography data after identification of rare membrane-associated and/or ribosome encapsulating Vaults within the dataset. The tomography and averaging approach appears to be technically sound, but the number of events identified is extremely small (14 for the membrane-associated Vaults, or ~1.5% of the total number of observed Vaults; 80 for the ribosome-encapsulated Vaults, or <10%) so the information gained from the structural analysis is very limited.

Some specific points:

1. Figure 1 panels E-I are all displayed at different apparent magnifications/levels of zoom, making comparison between the panels significantly more challenging.

We agree and have revised the panels as suggested. These are now part of the new Figure 1 (D-F). Similarly, we have applied this suggestion to all vault panels in the manuscript (Figs. 1, 2, 4).

2. The authors do not provide a figure of their proposed intact membrane-anchored Vault model (Figure 2D) docked into their subtomogram averaged map (Figure 2B), making it difficult for the reader to evaluate the fit of the model and interpret the apparent missing density away from the membrane (Figure 2).

We now provide the homology models we generated in the scope of the revisions fitted into our STA maps (both cytosolic and membrane bound vault STA) as ChimeraX sessions to the reviewers via the download link.

The authors also propose to deposit only the cytosolic Vault map from averaging in the EMDB, presumably due to issues around FSC/validation of the other maps due to the extremely limited particle numbers, which limits the ability for readers to perform this assessment on their own.

We have included a ChimeraX session into the submission to make the models transparent and will make them available upon publication using Zenodo.

3. The authors make repeated reference to part of the Vault structure being 'dissolved' in the membrane-associated form. I find this wording troublesome, given that they state that they do not observe this region of density in their maps or tomograms (eg. line 218-219) and therefore cannot unequivocally state what has happened to it. It is possible that the helical region associates with the membrane, which could easily be obscured by the membrane itself at this resolution as well as CTF effects, is dynamic or is even cleaved. A better choice of wording here might be to simply use 'unresolved', especially as the symmetry-equivalent part of the Vault away from the membrane is also poorly resolved in the map due to the low particle numbers.

We agree and have changed the wording to 'unresolved' throughout the text as suggested. We want to point out that the appearance of individual particles (Fig. 2D-G) and averages (Fig. 2C) is highly consistent and argues that a specific region of the vault remains unresolved. This region in MVP is the helical part residing C-terminal of the SPFH domain. It is unresolved only in the lower, but not upper half of the vault particle.

We consider CTF-effects unlikely because the upper half of the vault particle is resolved in the same subvolume. Furthermore, cleavage appears unlikely, as it remains undetected in Western blots (Fig. S2).

That the membrane itself obscures this part of the vault is however very much possible.

To take these potential effects into account, we have rephrased the text accordingly. The paragraph about individual events now reads as follows:

Lines 149-152 *"The part of the vault that is visible above the membrane accounted for approximately two thirds of the particle, while the part that is associated with the ER or NE (about one third) remained unresolved in the tomograms."*

... the paragraph about subtomogram averaging ...

Lines 175-180 *"While the part of the vault above the membrane remained intact, the cage-like density was not resolved underneath the bilayer density in the STA map, which is consistent with the primary data (compare Fig. 2C with Fig. 2D-G). Since both MVPs contain the alpha-helical domain and truncation of the protein was not observed by Western blotting (Fig. S2), this finding may imply that the membrane-facing part of the vault protein cage is structurally rearranged."*

... and the discussion:

Lines 396-400 *"We found that the association of vault particles with membranes occurred at a specific barrel height. Here, the alpha-helical domains forming one of the two vault caps remained unresolved in the EM maps, indicating that the cage may be rearranged from the SPFH domain onwards (Fig. 6C). In contrast, the lateral contacts formed by the MVP and SPFH domains between subunits were apparently largely preserved."*

4. In the single slice shown in Figure 2B of the membrane-associated Vault subtomogram average, there appears to be two patches of density below the membrane, at spacings that by eye could be consistent with the 'dissolved' Vault density. These features are also apparent in the map the authors provided for review. Given the density is comparable to that observed at the opposite end of the Vault, it needs careful consideration and at first glance could appear to be somewhat in contradiction with the authors' conclusions.

We had also observed this and were intrigued by this density because it occurred directly around the symmetry axis. We therefore repeated the subtomogram averaging procedure with lower initial low-pass filter settings. This procedure yielded

an improved map of the membrane-associated vault. In this average (new Fig. 2C) there is no strong density underneath the membrane, while other features remain the same.

In any case, given the low particle number of the membrane associated form, one has to be cautious with the structural interpretations other than the barrel height at which vaults associate with the membrane and the preservation of the upper part of the vault. We therefore have carefully edited the text to ensure that our conclusions are phrased cautiously enough.

5. I am hesitant about the interpretation of the connections between ribosomes and encapsulating vaults in Figure 3I, given the overall quality and SNR of the average which is apparent from the map provided to reviewers, with the vault itself being rather poorly resolved, most likely due to low particle numbers. As the authors propose to not deposit this map (I assume again due to FSC/validation issues) readers will be unable to make their own assessment of the interpretation. Likewise, interpreting a pattern in the orientation of the membrane-associated ribosomes from a sample of 4 individual events seems overly optimistic.

We would first like to clarify that we did not perform STA on the four ribosomes in vaults, but rather on all ribosomes of the dataset (Published dataset and analysis by Hoffmann et al.^{8,9}). This is indicated in the Methods section, where we note that we have used the orientations of ribosomes determined by this previous study.

In the revised version of the manuscript, we included statistical testing showing that the ribosome orientation within cytosolic vaults is significantly different from membrane-associated vaults. We refer to comment 6 to reviewer #1 for a detailed description on the statistical testing.

Importantly, we also provide independent support for the ribosome association of vault, as we identified a large set of ribosomal proteins in our TurboID proximity labelling experiment (new Fig. 5D).

Regarding the connecting density, we agree and have removed speculations about it from the text. Accordingly, the section now reads as follows (lines 280 ff):

“The ribosome translocon complex (RTC) facilitates translocation of ER lumen-bound proteins co-translationally. As part of the RTC, the 80S ribosome is positioned with respect to the ER membrane in a specific orientation and distance to allow insertion of the nascent chain into the translocon channel⁴³. In contrast to the vault-encapsulated ribosomes in the cytosol, three out of the four ribosomes observed inside membrane-bound vaults are positioned and oriented in a manner compatible with them being part of an RTC (hence referred to as RTC-like configuration). In this configuration, the longitudinal axes of ribosomes and vaults are roughly in register (Fig. 4H, K). We statistically compared the orientation of ribosomes contained within membrane-associated vault particles to those contained within cytosolic vault particles and found that it is significantly different ($p < 0.0001$ in a Fisher’s exact test, see methods for detail). However, the overall particle number is small and as such, this phenomenon should be further investigated in the future.”

6. Given the extremely small number of membrane-associated/ribosome-encapsulating Vaults, a bigger cryoET dataset showing Vault and membrane/ribosome co-localisation/association would improve many of the issues described above and add significant weight to the robustness and importance of the authors' observations. An orthogonal experimental technique or model organisms could also help towards validating the observations.

We agree that the number of captured events is low for membrane-associated and ribosome-encapsulating vaults. Thus, as suggested, we now provide additional experimental support from a proximity labelling experiment, where we fused the biotin ligase to the C-terminus of MVPA. In this experiment, we treated cells with biotin for 15min, performed a streptavidin pulldown and compared the biotinylated proteins to a control, where the biotin ligase is cytosolic. We find that ribosomal and ER proteins are part of the vault-associated proteome (new Fig. 5), thus independently confirming our structural data. The identification of ER-resident (luminal) proteins also implies that the C-terminus of MVP reaches into the ER lumen, but we do not speculate about this in the manuscript. In addition, we now also assess the membrane characteristics at the site of vault association and find that these are different than the surroundings (new Fig. 3).

References

1. John Travis. The vault guy. <https://www.science.org/content/article/biologist-aims-solve-cell-s-biggest-mystery-could-it-help-cancer-patients-too> (2024).
2. Glushkova, D., Böhm, S. & Beck, M. Systematic membrane thickness variation across cellular organelles revealed by cryo-ET. *J Cell Biol* **225**, (2026).
3. van Kempen, M. *et al.* Fast and accurate protein structure search with Foldseek. *Nat Biotechnol* **42**, 243–246 (2024).
4. Collins, B. M. Revealing the architecture of the membrane-bound Flotillin cage assembly. *Proceedings of the National Academy of Sciences* **121**, (2024).
5. Li, H., Vallese, F. & Clarke, O. B. The vault particle is enclosed by a C13-symmetric cap with a positively charged exterior. Preprint at <https://doi.org/10.1101/2025.06.06.658390> (2025).
6. Daumke, O. & Lewin, G. R. SPFH protein cage — one ring to rule them all. *Cell Research* vol. 32 117–118 Preprint at <https://doi.org/10.1038/s41422-021-00605-7> (2022).
7. Ma, C. *et al.* Structural insights into the membrane microdomain organization by SPFH family proteins. *Cell Research* 2021 32:2 **32**, 176–189 (2022).
8. Hoffmann, P. C. *et al.* Structures of the eukaryotic ribosome and its translational states in situ. *Nat Commun* **13**, 7435 (2022).
9. Hoffmann, P. C. *et al.* Nuclear pore permeability and fluid flow are modulated by its dilation state. *Mol Cell* **85**, 537-554.e11 (2025).
10. Kowalski, M. P. *et al.* Host Resistance to Lung Infection Mediated by Major Vault Protein in Epithelial Cells. *Science* (1979) **317**, 130–132 (2007).
11. Yang, J. *et al.* Vaults are dynamically unconstrained cytoplasmic nanoparticles capable of half vault exchange. *ACS Nano* **4**, 7229–40 (2010).
12. Chugani, D. C., Rome, L. H. & Kedersha, N. L. Evidence that vault ribonucleoprotein particles localize to the nuclear pore complex. *J Cell Sci* **106**, 23–29 (1993).

13. Kedersha, N. L., Miquel, M. C., Bittner, D. & Rome, L. H. Vaults. II. Ribonucleoprotein structures are highly conserved among higher and lower eukaryotes. *J Cell Biol* **110**, 895–901 (1990).
14. Mrazek, J. *et al.* Polyribosomes Are Molecular 3D Nanoprinters That Orchestrate the Assembly of Vault Particles. *ACS Nano* **8**, 11552–11559 (2014).
15. Buehler, D. C. *et al.* Bioengineered Vaults: Self-Assembling Protein Shell–Lipophilic Core Nanoparticles for Drug Delivery. *ACS Nano* **8**, 7723–7732 (2014).
16. Zheng, C.-L. *et al.* Characterization of MVP and VPARP assembly into vault ribonucleoprotein complexes. *Biochem Biophys Res Commun* **326**, 100–7 (2005).
17. Vasu, S. K. & Rome, L. H. Dictyostelium Vaults: Disruption of the Major Proteins Reveals Growth and Morphological Defects and Uncovers a New Associated Protein. *Journal of Biological Chemistry* **270**, 16588–16594 (1995).
18. Vasu, S. K., Kedersha, N. L. & Rome, L. H. cDNA cloning and disruption of the major vault protein α gene (*mvpA*) in Dictyostelium discoideum. *Journal of Biological Chemistry* **268**, 15356–15360 (1993).
19. Guerra, P. *et al.* Symmetry disruption commits vault particles to disassembly. *Sci Adv* **8**, 7795 (2022).

Reviewer #1 (Remarks to the Author):

I have re-reviewed the revised manuscript "The vault associates with membranes in situ" and I have read the detailed rebuttal letter describing how the authors have responded to all of the reviewer comments.

I think that the authors have done an excellent job responding and re-submitting a much more convincing manuscript that contains considerable new data and additional analysis of their previous data.

Although this manuscript does not solve the mystery of vault function, there are two significant findings in the manuscript that should contribute new insights to the field. I certainly agree with the authors statement in the rebuttal letter that their finding "that a minor fraction of vault particles associates with membranes at a defined barrel-height, and that it encapsulates ribosomes in preferred orientations in situ" is indeed novel and potentially important for defining additional functional experiments.

I have no additional criticisms.

We thank the reviewer for the constructive comments, helpful suggestions and the positive assessment of our work.

Reviewer #2 (Remarks to the Author):

My assessment of this manuscript is not altered by the revision. The main finding of the manuscript, the identification and low-resolution structural characterization of membrane-bound vaults, is a noteworthy and surprising discovery, but without further functional or structural evidence, the relevance and the topology of the membrane-embedded vault remains enigmatic and the manuscript appears too preliminary for Nat Comm. I have also some concerns about the revision.

We are sorry that we could not convince the reviewer about the relevance of our findings for the vault field and the scientific community. As we have repeatedly acknowledged, we do not claim to solve the mystery surrounding the vault's cellular function, including its membrane association. However, we are convinced that the findings we report are an important step towards this goal, as they can be used to design future experiments in a directed manner. Thus, while we do delineate a full mechanism or function, we respectfully disagree with the statement that it is overall too preliminary for publication.

Figure 2: I indeed wrongly assumed in my first review that, similar to other SPFH members, a half capsule of the vault, not a full vault, binds to cellular membranes. This misunderstanding is partially caused by insufficient structural presentation and labelling, which has not been improved in the revised version: Based on the subtomogram average in Fig. 2C, it is not so obvious that a full vault particle is embedded in the membrane, and the reader should not wait until Fig. 6 to understand the main findings of the paper (see also point 2 of referee 3). To avoid confusion, Fig. 2 should already contain a structural model of membrane-bound vault

in which the individual domains and the different regions of vault (e.g. cap, shoulder, etc) are clearly labelled and the molecules of the two hemi-capsules, as far as they are visible, are displayed in different colours. Showing a half-capsule of the vault (previously in Fig. 2, now in Fig. 6B) next to the related half cages of other SPFH members is also not adding clarity in this regard.

We thank the reviewer for these comments and further clarifying the issue. To make our point more obvious to the reader, we have revised the text as follows:

lines 150-52 *“The part of the vault that is visible above the membrane accounted for approximately two thirds of the particle, thus not representing a vault half-cage. In contrast, the part that is associated with the ER or NE (about one third) remained unresolved in the tomograms.”*

lines 181-84 *“While the part of the vault above the membrane remained intact, the cage-like density was not resolved underneath the bilayer density in the STA map, which is consistent with the primary data (compare Fig. 2C with Fig. 2D-G). Since both halves of the vault particle contain the alpha-helical domain of the MVP, and truncation of the protein was not observed by Western blotting (Fig. S2), this likely reflects incomplete visualization of the membrane-facing region of an otherwise intact vault cage and may imply that this part of the vault protein cage is structurally rearranged.”*

As suggested, we have also added a second half-cage to Figure 6B. We would prefer to not show a structural model already in Figure 2, in order to clearly separate experimental data from interpretation. Instead, we marked the equatorial plane in Figure 2 to make it more obvious that both halves are present.

Figure 2: With a full vault embedded in the membrane, its membrane-interaction mode is even more dazzling. Do the authors envision that the entire cap structure of the partially resolved hemi-capsule is post-translationally threaded through the membrane, with the MVP domains acting as transmembrane-spanning regions (as implied in the subtomogram averages of membrane-associated vaults in Fig. 2C, see point 4 of referee 3)? Such threading seems energetically costly, if not impossible. If the vault was such a membrane-traversing complex, how could then the membrane enter into the interior of the capsule? Alternatively, is the non-resolved half-capsule rather a peripheral membrane-binding disc, similar to caveolin, that can bind to membranes and be released without extraction of the transmembrane regions?

We agree that these are very interesting questions that could now be addressed based on our findings. However, the reviewer is asking us to speculate about aspects that are beyond the scope of our study and we would like to refrain from doing so in the manuscript.

Fig. 3: Along the same line: The authors determine ‘membrane thickness’ within the vault-encapsulated) membrane but a major portion of the vault protein present in this area is not resolved.

How can the authors ascertain that the ‘membrane’ density in the interior of the vault does not represent the unresolved protein density (for example, in the form of a disc,

see again for the caveolin structure rather than the membrane; and that difference in 'membrane thickness' may be related to the presence of protein rather than differences in bilayer thickness? How should the vault select for short-chain lipids yielding a thinner membrane in its interior if not via protein-lipid interactions?

We do not believe that the membrane density in the interior of vault reflects the unresolved protein density. The algorithm we use to measure membrane thickness includes an intrinsic measure for quality- intensity plots of bilayer density. For bilayer membranes, these should have two minima (phosphate-rich headgroup regions) with a central maximum between them (hydrophobic core)^{1,2}. As a matter of fact, we have recently shown that the density line profiles we observe here (Fig. 3B) are characteristic of lipids, this does not exclude that protein may be present in the enclosed patch². We therefore conclude that the measured region includes membrane and not only protein.

In addition, we did not intend to claim that vault is actively selecting for a certain lipid type in its interior. At this stage, we do not know if the vault is recruited to certain membrane regions, which are thinner, for example due to a specific lipid or composition, or if binding of the vault to membranes leads to a thinning or remodeling of membranes. All we can say from our data is that the membrane enclosed by the vault is thinner than the membrane on the outside of the particle in indirectly adjacent regions. Again, we would like to refrain from speculating about this any further.

If the authors cannot provide further cell-based data to characterize the function of membrane-embedded vaults, they could attempt to characterize its membrane-binding determinants in terms of lipid composition and membrane curvature *in vitro*. Furthermore, they could reconstitute vault in suitable detergent or membranes and determine a higher resolution structure, in which its correct topology is determined and some additional insights into the formation of such complex are obtained. Such experiments would make this study suitable for Nat. Comm.

We thank the reviewer for these suggestions. However, establishing, optimizing and carrying out such experiments would take a significant amount of time (months to year(s)) and it is not clear if such an *in vitro* set-up would be able to recapitulate the situation in cells accurately enough. For example, vault binding to membranes may be driven by specific cellular factors, modifications or regulatory cues, which would be absent *in vitro*. In addition, even if successful, the results would not add much to clarifying vault's function at the membrane. Thus, while being interesting aspects for potential future structural studies, we believe that addressing them is beyond the scope of this revision.

Further comments:

Line 280 et seqq.: Vault and ribosome translocon complex, orientation of the ribosomes within membrane-bound vault.

I do not understand these arguments. Were the ribosome-encasing, membrane-bound vaults found on ER membranes where a translocon may be present, or rather at nuclear membranes? Is there any hint that the translocon is associated with vault complexes? To me, these ideas appear vastly speculative, and are also not

supported by the low number of relevant particles; if at all, such suggestions should be included in the discussion.

We first want to clarify that the translocon is known to localize to both, the ER and the outer nuclear membranes. However, our intention in that section was to make clear in which orientation the ribosome is inside of the vault: the ribosome and its exit tunnel are oriented such that they could be part of a translocon complex at the ER membrane, where the nascent chain from the ribosome is channeled into the ER lumen – and this is statistically unlikely to occur by chance (Figure 4). We do not directly detect a full translocon complex at the sites of vault membrane binding in our data.

Reviewer #3 (Remarks to the Author):

The changes the authors have made have significantly improved the manuscript. The addition of the mass spectrometry/proximity labeling in particular provides much needed orthogonal methods.

We appreciate the reviewer's positive assessment.

The authors have addressed the first set of points satisfactorily. A few small comments:

1) It would be useful to have some sort of indication of how many vaults showed non-ribosomal density inside the cavity, in order to put the ribosome-containing vaults in context

The vast majority of vaults contains multiple heterogeneous densities inside (exemplified in Figure 1D-F). Those features are typically much smaller as compared to ribosomes. We did not find any obvious repetitive, or well-defined patterns across particles, making it difficult to classify and quantify reliably.

2) Some of the figures with tomogram slices suffer from uneven contrast/processing (in particular Fig 1 E-I, Fig 2 D-G) which detracts from the interpretability of the figures.

We appreciate the reviewer's comment and we have revised the tomogram slices in Fig. 1D-F and Fig. 2D-G to improve visualization. This included adjusting contrast and histogram levels to enhance interpretability. However, despite these adjustments, some differences in contrast do remain. These reflect inherent variability in the data that were initially acquired in different osmotic conditions^{3,4} and therefore have different cytosolic protein densities.

3) In Figure 3A, on the right side panel, it is hard to see the horizontal coloured stripes on top of the orange to blue density colouring. Alternate colours or a greyscale density background would make this easier to see.

As suggested, we have changed the colour scheme as well as the line thickness.

4) The particle count on Figure 4H should only show integer values - as its a count, non-integer values are confusing.

We agree with the reviewer and have adapted the scale accordingly.

References

1. Falck, E., Patra, M., Karttunen, M., Hyvönen, M. T. & Vattulainen, I. Lessons of slicing membranes: Interplay of packing, free area, and lateral diffusion in phospholipid/cholesterol bilayers. *Biophys. J.* **87**, 1076–1091 (2004).
2. Glushkova, D., Böhm, S. & Beck, M. Systematic membrane thickness variation across cellular organelles revealed by cryo-ET. *J. Cell Biol.* **225**, (2026).
3. Hoffmann, P. C. *et al.* Nuclear pore permeability and fluid flow are modulated by its dilation state. *Mol. Cell* **85**, 537-554.e11 (2025).
4. Hoffmann, P. C. *et al.* Structures of the eukaryotic ribosome and its translational states in situ. *Nat. Commun.* **13**, 7435 (2022).